# The dynamic distribution of genetic tandem amplifications in a heteroresistant *Escherichia coli* population revealed by ultra-deep long read sequencing

Sofia Jonsson [1], Andrei Guliaev [1,4], Brandon A. Berryhill [2,3,5],
Dan I. Andersson [1] & Hervé Nicoloff [1] ✉

Antibiotic heteroresistance, characterized by rare resistant subpopulations of bacteria within a susceptible main population, is associated with treatment failure and often caused by tandem amplification of resistance genes. Here, we investigated how the distribution of tandem amplifications affects hetero-resistance using an approach combining genetic engineering and ultra-deep Nanopore sequencing to accurately quantify the distribution of tandem amplification copy numbers on plasmids down to frequencies of $10^{-5}$. Using an *Escherichia coli* isolate, we describe the direct relation between the distribution of tandem amplifications increasing the copy number of a $bla_{SHV}$ gene and a heteroresistance phenotype to piperacillin-tazobactam, and reveal how this distribution expands under antibiotic pressure and partially reverts upon its removal. Mathematical modeling indicates that indirect resistance and fitness cost of amplifications influence the dynamic distribution of tandem amplifications. These findings provide insights into amplification-mediated phenotypes and enhance possibilities for the development of improved therapeutic and diagnostics strategies for heteroresistance.

Antibiotic resistance is a known threat to successful treatment of bacterial infections, estimated to be associated with 4.95 million deaths annually[1]. Detection of resistance and understanding the molecular mechanisms that causes it are important parts of tackling the problem[2]. Heteroresistance (HR) is a phenotype where low frequency subpopulations with increased antibiotic resistance are present within a main susceptible population of bacteria[3–5]. Since both animal models and clinical studies show an increased risk of treatment failure for drugs against which the infecting pathogen is heteroresistant[6–11], there is a need for a deeper understanding of the mechanisms behind this phenotype. However, the low frequency and instability of the resistant subpopulation cause specific challenges regarding both genetic investigation and HR detection.

Spontaneous genetic tandem amplification of resistance-conferring genes is a common mechanism of HR in Gram-negative pathogens[4]. These can increase the copy number of genes in pathways associated with the drug target[12,13] or of antibiotic resistance genes[4,14,15]. For genes that cause low levels of resistance when present at a single copy, the increased copy number following tandem amplification leads to increased resistance, sometimes reaching clinical breakpoint[4,16]. Due to their genetic instability and fitness cost, tandem genetic amplifications often have highly dynamic characteristics, and

[1]Department of Medical Biochemistry and Microbiology, Uppsala University, Uppsala, Sweden. [2]Department of Biology, Emory University, Atlanta, GA, USA. [3]Program in Microbiology and Molecular Genetics, Graduate Division of Biological and Biomedical Sciences, Laney Graduate School, Emory University, Atlanta, GA, USA. [4]Present address: Clinical Genomics Uppsala, Science for Life Laboratory, Uppsala University, Uppsala, Sweden. [5]Present address: Department of Medical Biochemistry and Microbiology, Uppsala University, Uppsala, Sweden. ✉e-mail: herve.nicoloff@imbim.uu.se

the frequency of resistant cells can rapidly change in response to altered selection pressure[9,17–20]. This instability likely affects detection of tandem amplifications in clinical samples. Yet, they have been observed in some patients where they were associated with resistance development and treatment failure[21–23]. Indirect evidence linking tandem amplifications and treatment failure in patients also exists. For example, HR to piperacillin-tazobactam (TZP) in *Escherichia coli* bloodstream infection isolates has been linked to both tandem amplifications of β-lactamase genes located on plasmids in vitro[16], and to increased risks of antimicrobial treatment failure in patients[7].

A better understanding of the dynamic distribution of amplification copy numbers (ACN) during growth of bacterial populations is needed to better understand the nature of the HR phenotype and how different antibiotic selection pressures affect the rise and decline of the resistant cells in the population. However, commonly used detection methods do not allow for precise and direct investigation of tandem genetic amplifications at the single cell level within a population. For example, while incorporation of a fluorescent reporter gene in the amplification unit enables for single cell amplification copy number detection[24], the reporter gene alters the original genetic context and might affect the fitness cost of amplifying the genetic unit, potentially disturbing the dynamic distribution of tandem amplifications in the population. Furthermore, non-linear scaling of fluorescence and signal overlap limit the accuracy of single-cell measures, especially as ACNs increase[24]. Genetic methods of ACN detection, including short read whole genome sequencing (WGS), quantitative PCR (qPCR), and digital droplet PCR (ddPCR), only quantify the average copy number of tandem amplifications present in the population and lack the single-cell level of detection required for analyzing the distribution of tandem amplifications in populations[17,24]. Besides, when tandem amplifications are rare these methods fail at detecting their presence. Therefore, to achieve single-cell genetic resolution, new methods are needed with sufficient throughput to capture the low frequency tandem amplifications that are expected in subpopulations of HR isolates. Such methods should not genetically alter the amplified unit as this could affect its frequency and distribution. Nanopore long read sequencing has promising potential as it allows for the detection and precise quantification of full arrays of tandem amplified units present on single DNA molecules. Accurate Nanopore-based quantification of all amplified units that are present on a DNA molecule requires sequencing reads covering the DNA sequences located upstream and downstream of the amplified region, as these mark the beginning and end of the array of amplified units. Nanopore sequencing was used, for example, to reveal a diversity of ACNs within colonies of resistant bacteria, or to confirm the presence of tandem amplifications in a clinical isolate[22,25]. Unfortunately, studies relying on Nanopore sequencing had low sequencing depths covering the entire amplification arrays, which prevents the analysis of both low-frequency events and the distribution of amplified units[22,25–28]. This demonstrates the challenge of achieving the sufficient sequencing depth and read length required to detect tandem genetic amplifications in subpopulations of bacteria.

In this study, we describe an approach that relies on ultra-deep Nanopore long read sequencing for the detection of full amplification arrays present on plasmids, and which allows for detection of individual arrays present at frequencies as low as $10^{-5}$. We use the method to assess how the distribution of antibiotic resistance-conferring tandem amplifications, present on a clinical plasmid in a *E. coli* strain showing a TZP HR phenotype, is affected by growth with and without antibiotics and impacts the HR phenotype. *E. coli* is a major cause of urinary tract infections and bloodstream infections[29] and amplification-mediated HR in *E. coli* is frequently plasmid-associated[4,9,16]. Our results reveal previously unknown insights into the dynamic distribution of tandem genetic amplifications and the factors such as selection pressure, fitness costs and indirect resistance that shape them in bacterial populations.

## Results

### Genetic engineering for optimized detection of tandem amplifications

To optimize the detection of tandem amplifications by Nanopore sequencing, we genetically modified a plasmid from a TZP HR clinical *E. coli* isolate that we conjugated into *E. coli* MG1655. This 116 kbp plasmid, pDA61218_116, is present at a single copy per cell in the original isolate DA61218 and has a SHV-1 β-lactamase located in a 3.5 kb amplification unit flanked by IS*26* elements present in the same orientation (Fig. 1a). We integrated a unique I-*Sce*I restriction site in a neutral locus 1659 nucleotides downstream of the amplification array, with no other I-*Sce*I sites present in the genome. Population analysis profile (PAP) test confirmed that the TZP HR phenotype of the strain carrying the modified plasmid (DA76595) was similar to that of the original clinical isolate DA61218 (Supplementary Fig. 1). To increase sequencing coverage over the amplification array, plasmid DNA (pDNA) was extracted, linearized by I-*Sce*I restriction digestion, and Nanopore sequenced. Sequencing of pDNA rather than of whole genomic DNA (gDNA) increases the sequencing depth by a theoretical 41-fold by preventing unnecessary sequencing of the chromosomal DNA (when comparing sequencing of the 116 kbp plasmid alone versus sequencing of both the plasmid and the 4649 kbp genome of *E. coli* MG1655). I-*Sce*I digestion allows for the targeted generation of sequencing start sites located close to the amplified unit, and for sequencing of circular plasmids present in the pDNA extractions. We evaluated the impact of the I-*Sce*I site on the sequencing by comparing sequencing depths with and without I-*Sce*I digestion. For I-*Sce*I-digested pDNA, the sequencing depth of the region carrying the amplified unit was increased 22% compared to that of a control region located at the opposite side of the plasmid (Fig. 1d). Sequencing of two pDNA extractions that were either undigested or digested with I-*Sce*I resulted in 19 out of 1,571,879 (0.001%) and 5824 out of 1,709,468 (0.3%) reads covering the whole plasmid length, respectively.

### Distribution of tandem amplification copy numbers in a HR isolate grown in absence of antibiotic selection pressure

We hypothesized that the pDA61219_116-dependent HR phenotype results from subpopulations of bacteria with varying copy numbers of spontaneous tandem amplifications of $bla_{SHV}$. To determine whether such subpopulations exist, strain DA76595 was cultured in MH broth in absence of antibiotics in five independent replicates (DA79881, DA79882, DA79883, DA81276 and DA81279), and I-*Sce*I-digested pDNA was sequenced using a full Nanopore flow cell per sample. In parallel, gDNA was extracted from the same five cultures and analyzed using two independent methods detecting the average copy number of the amplified unit: short read WGS and ddPCR. It is important to note that the plasmid pDA61218_116 carries a duplication of the amplified unit, which was present in the original clinical isolate. Analysis of the Nanopore sequencing reads (see "Methods", Supplementary Fig. 2) revealed the presence of varying copy numbers of amplified units in the bacterial cells, with ACNs ranging from 0 to 14 detected at frequencies as low as $10^{-5}$ (Fig. 2, Supplementary Data 1). To validate our Nanopore data, we controlled that the average copy number of amplified units detected by Nanopore sequencing of pDNA was similar to that detected by short reads WGS and ddPCR of gDNA. All three methods showed agreement, with an average of 2 amplifications per plasmid for the five analyzed cultures (Supplementary Table 1), indicating that Nanopore sequencing did not, on average, over- or underestimate the ACNs. WGS and ddPCR confirmed the presence of pDA61218_116 at an average plasmid copy number (PCN) of 1.15 ± 0.34 copy per cell (Supplementary Table 1).

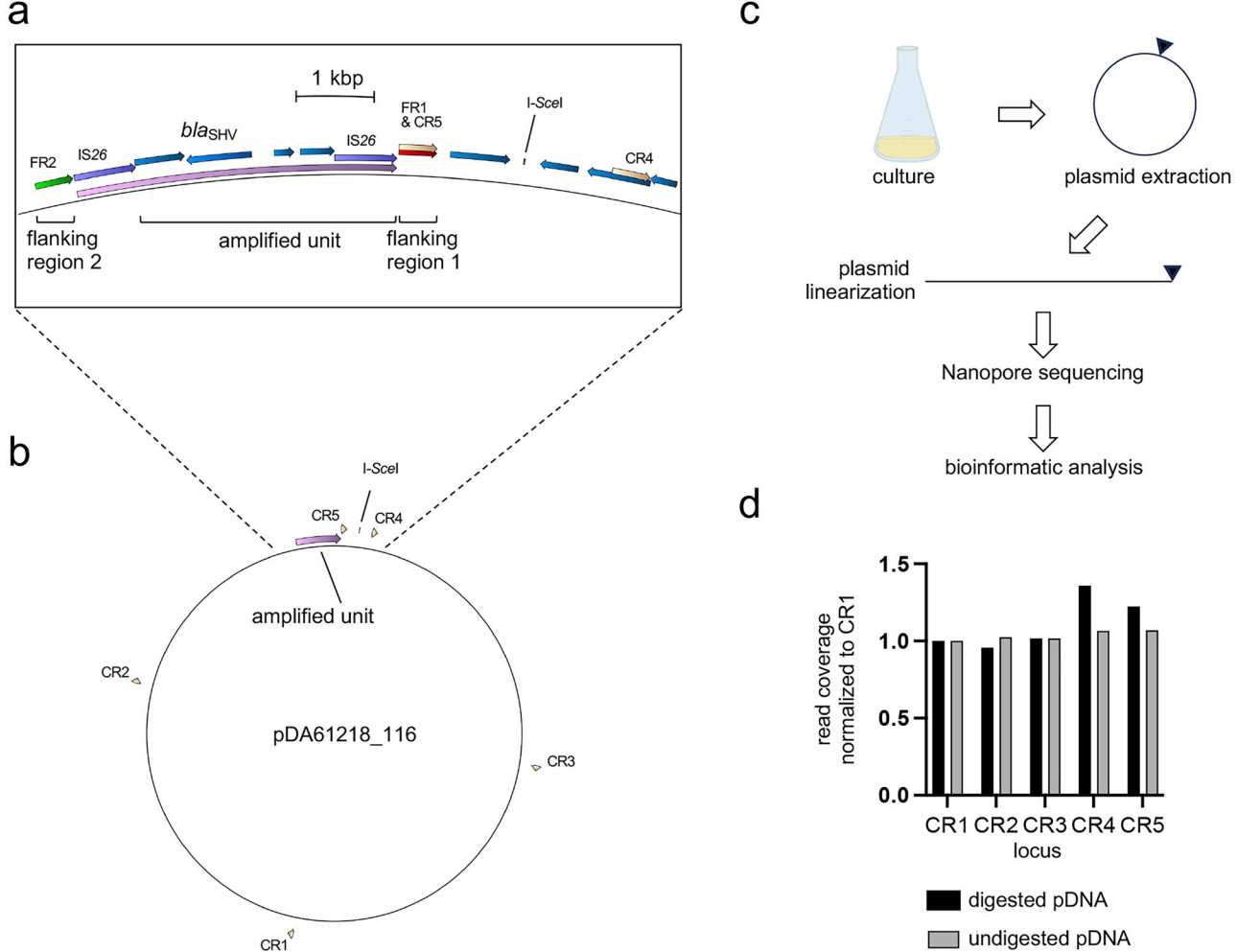

**Fig. 1 | Experimental overview. a** Genetic context around the amplified unit. The repeated sequences (IS*26*) surrounding the amplified unit encoding *bla*$_{SHV}$, and the regions that are important for Nanopore sequencing and bioinformatic analysis (I-*Sce*I restriction site, flanking regions FR1 and FR2, and control regions CR4 and CR5) are shown. **b** Plasmid pDA61218_116 and location of the amplified unit, I-*Sce*I restriction site, and the control regions CR1 to CR5. **c** Outline of workflow for Nanopore sequencing. The I-*Sce*I restriction site is represented by a black triangle. **d** Nanopore sequencing read coverage comparison over five plasmid control regions (CR1 to CR5; see **b**). Sequencing coverages were normalized towards coverage at the CR1 locus. Results for sequencing of pDNA that was either digested (dark bars) or not (light bars) by I-*Sce*I prior to Nanopore library preparation are shown.

## The amplification copy number distribution is dynamic and affected by the growth conditions

We then proceeded to assess how exposure to antibiotics affects the distribution of tandem amplifications in the HR population. For this, four of the independent replicates described above were transferred to broth growth medium supplemented with 3 mg L$^{-1}$ TZP, a concentration allowing for growth of the whole population according to PAP tests results (Supplementary Fig. 3). The cultures were then further transferred to broth supplemented with increasing antibiotic concentrations: two cultures were transferred to 6 and then 9 mg L$^{-1}$ TZP (3 mg L$^{-1}$ increments, replicates DA81276 and DA81279), and the other two to 6 and then 12 mg L$^{-1}$ TZP (2-fold increments, replicates DA79881 and DA79882). After each incremental step, we investigated the ACN distribution as described above, but multiplexed 12 samples per flow cell for Nanopore sequencing. For all four replicates, no significant difference (two-sample Kolmogorov Smirnov test, Supplementary Table 2) was detected when the distribution was compared to the previous culture condition, up to 6 mg L$^{-1}$. However, for the highest TZP concentrations, the distribution significantly changed (two-sample Kolmogorov-Smirnov test, Supplementary Table 2). The proportion of higher ACNs increased while still maintaining populations with lower copy numbers (Fig. 3, Supplementary Figs. 4 and 5). Copy numbers between 1 and 19 could be detected for selection at 9 mg L$^{-1}$, and between 1 and 20 for selection at 12 mg L$^{-1}$, down to a frequency of 10$^{-4}$.

We then investigated the stability of these distributions when the populations were grown in absence of antibiotics. For this, we transferred the four independent cultures grown in TZP 9 or 12 mg L$^{-1}$ to MH broth without TZP and grew them for an additional 320 generations (32 passages). The ACN distributions reverted relatively slowly (Fig. 3, Supplementary Figs. 4 and 5). For the two cultures selected at 12 mg L$^{-1}$ TZP, reversion was slower than for the two cultures selected at 9 mg L$^{-1}$ TZP (two-sample Kolmogorov-Smirnov test, *D*-values of 0.72 and 0.70 for cultures in 12 mg L$^{-1}$ TZP and 0.10 and 0.13 for cultures in 9 mg L$^{-1}$ TZP, respectively, when comparing the initial distribution in MH broth versus the distribution after reversion for 320 generations; Supplementary Table 2).

For all timepoints, WGS and ddPCR confirmed the presence of pDA61218_116 at an average PCN of 1.01 ± 0.23 copy per cell (Supplementary Table 1). These results showed that, throughout the whole experiment, a single copy of the plasmid carrying a specific copy number of amplified units was present, on average, in each bacterium.

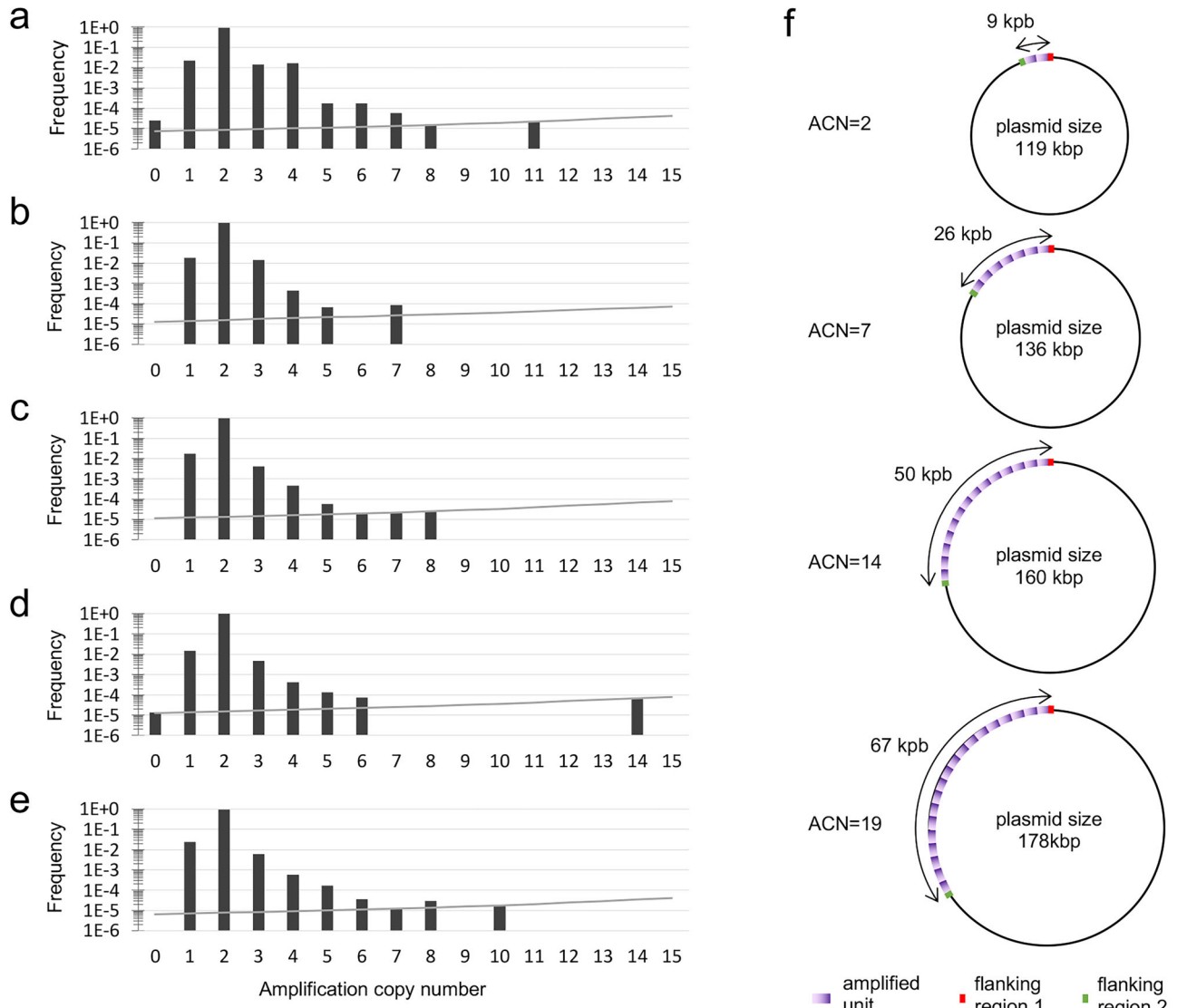

**Fig. 2 | Amplification copy number distribution in broth without antibiotics detected through Nanopore sequencing of linearized plasmid DNA.** Five independent replicates of DA76595. Bars indicate amplification copy number distribution, and lines indicate the sequencing limit of detection. **a** Replicate DA79881; **b** Replicate DA79882; **c** Replicate DA79883; **d** Replicate DA81276; **e** Replicate

DA81279. **f** Examples of plasmids carrying different amplification copy numbers (ACN). The minimum size of the Nanopore sequencing reads that can sequence the whole arrays of amplified units including the unique flanking regions 1 and 2 present on each side of the arrays of amplified units are indicated next to the amplified units.

However, we could not exclude the possibility of subpopulations of bacteria carrying increased plasmid copy numbers, and therefore we conservatively concluded that our tandem ACN were accurately detected at single-molecule level rather than single-cell level.

For ACNs, agreement between Nanopore sequencing of pDNA and short reads WGS and ddPCR analysis of gDNA varied from 0.95 (no difference) to 1.83-fold change (1.83-fold higher average ACN detected by WGS and ddPCR than by Nanopore sequencing). For a majority of the conditions (37 out of 45 sequenced culture conditions), agreement was between 0.95 and 1.4 (Supplementary Table 1). To determine if the detected discrepancies resulted from differing DNA extraction efficiency between gDNA and pDNA, we analyzed by ddPCR a subset of timepoints using DNA templates extracted by bead-beating, a mechanical extraction method that accurately extracts the whole bacterial genetic content[30]. Similar average ACNs were detected with bead-beating and gDNA extractions (Supplementary Table 4), indicating that the gDNA extractions did not introduce any bias in the average ACN detected, and that pDNA extractions or Nanopore

sequencing sometimes led to lower average ACN copy numbers than expected. To determine if the lower ACN detected was caused by the pDNA extractions themselves or by the Nanopore library preparation or sequencing, ddPCR analysis was performed on a subset of pDNA extractions that had also been used for Nanopore sequencing. Good agreement between ddPCR and Nanopore sequencing was observed (agreements of 0.95 to 1.05; Supplementary Table 3). This indicated that the Nanopore library preparation and sequencing did not affect the average ACNs detected, which was instead caused by the pDNA extractions less efficiently extracting plasmids with higher copies of tandem amplifications. When analyzing data in the Supplementary Table 1, we observed a strong positive correlation between the average ACN detected in the gDNA extractions and the fold difference between average ACN detected in gDNA versus Nanopore sequencing (Pearson correlation analysis, $R = 0.9111$, $p < 0.0001$). Thus, the discrepancies observed were consistent and the pDNA extractions introduced a bias in our results, which increased with the average ACN[30].

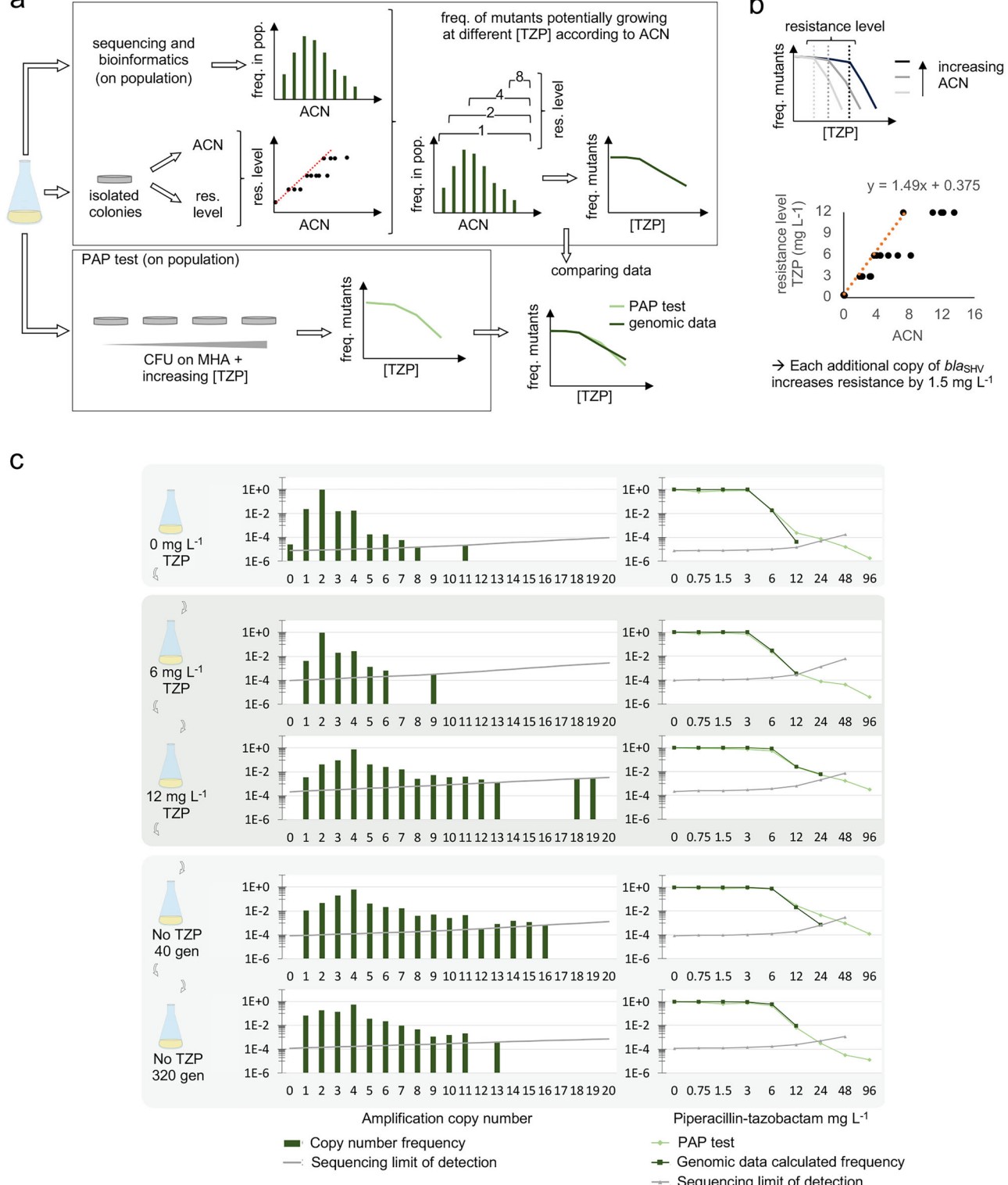

**Fig. 3 | Dynamics of amplification copy number distribution and comparison of the populations HR phenotype and genetic distribution. a** Schematic representation of the experiment. Res resistance, TZP piperacillin-tazobactam. **b** Increased TZP resistance level per each additional copy of $bla_{SHV}$. The resistance level of colonies with different ACNs was determined as the highest concentration of TZP allowing growth of ≥20% of the population. **c** Correlation between PAP test and ACN. On the left are frequencies (y axis) of ACNs detected by Nanopore sequencing (bars) and the limit of detection of each ACN (gray line). On the right are frequencies of resistant subpopulations detected by PAP test (pale green) and frequencies of resistant subpopulations that should grow at the different TZP concentrations according to Nanopore ACN distribution data (dark green). Gray line: Nanopore sequencing limit of detection of the lowest ACN required for growth. Data shown is replicate DA79881.

## Correlation between copy number distribution and HR phenotype

We then assessed how well the ACN distribution correlated with the populations' HR phenotype. For this, eight of the cultures of replicate DA79881 were investigated. PAP tests were performed to determine the frequency of the subpopulations of bacteria that can grow at specific TZP concentrations. To estimate the resistance level associated with each additional copy of $bla_{SHV}$, we isolated colonies carrying different copy numbers of the amplified unit (as determined by ddPCR on DNA templates prepared by bead-beating), and measured a TZP resistance level increase of 1.5 mg L$^{-1}$ per additional copy (Fig. 3b). We then compared the frequencies of subpopulations detected in the PAP tests to the frequencies of bacteria carrying enough copies of $bla_{SHV}$ to allow growth in presence of the TZP concentrations of the PAP tests (Fig. 3a). Very good agreement was observed (Supplementary Fig. 7), and cultures with higher ACNs led to higher frequencies of subpopulations growing in presence of higher TZP concentrations (Fig. 3c, Supplementary Fig. 7). As expected, frequencies of tandem amplifications calculated from ≤2 sequencing reads or close to the Nanopore sequencing limit of detection led to more variability in agreement (Supplementary Fig. 7, Supplementary Data 1). Thus, HR was due to subpopulations carrying different copy numbers of tandem genetic amplifications and was affected by their distribution in the population.

## Media adaptation mutations during prolonged growth

Media adaptation mutations can occur during continuous broth culturing[31,32]. To evaluate for this, short read WGS data was analyzed for presence of background mutations in all timepoints of the cultures investigated (Supplementary Data 2). In all four replicates, mutations, mostly on the chromosome (29/30), appeared at different timepoints during growth. A majority (27/30) accumulated during the 320 generations of growth in absence of antibiotics, and 21/27 were located in genes or pathways previously described to confer media adaptation. Only one mutation, appearing during growth in presence of TZP, might confer a phenotypic advantage in presence of antibiotics; a mutation in $wecH$ linked to antibiotic tolerance[33].

Since these additional mutations occur in bacteria carrying plasmids with specific copy numbers of tandem amplifications, and since we observed that tandem amplifications were relatively stable over time, we expected that these mutations would impact the distribution of ACNs, with specific ACNs becoming over-represented as mutants become dominant in the populations. Indeed, we observed such phenomena (Supplementary Fig. 8). For example, a $wecH$ mutation selected in replicate DA79881 in presence of TZP led to a population dominated by ACN = 4, while the other replicates did not accumulate additional mutations and had wider ACN distributions. Further accumulation of mutations increasing fitness ($arcA$, $trkH$ and $malS$) in the $wecH$ population led to a population with ACN = 4 still dominating replicate DA79881 after 320 generations of growth in absence of TZP. Similarly, in replicate DA79882, an $arcA$ mutation positively selected a population with ACN = 5, while in replicate DA81276 the fitter mutants positively selected a population with ACN = 2. Thus, by positively selecting populations with specific ACNs, fitter mutants appearing during growth impacted the ACN distributions, which could slow down or accelerate ACN reversion in absence of antibiotics.

Importantly, when no new media adaptation mutations appeared over numerous generations of growth in absence of TZP, the ACN distributions remained relatively stable (Supplementary Fig. 8). For example, during 80 generations of growth of replicate DA79881 where only a $wecH$ mutation was detected, the ACN distribution reverted slowly. Similarly, for replicates DA79882 and DA81276, ACN distributions remained relatively stable for 40 and 80 generations, respectively, until the first media-adapted mutants appeared and increased in proportion in the population, thereby disturbing the ACN distribution.

Finally, during growth of DA81279, no clone with background mutation was successful enough to dominate the population (Supplementary Data 2). Yet, even after prolonged growth in absence of TZP, ACN reversion was incomplete. For example, after 320 generations of growth in absence of antibiotics, populations with ACN = 3 and ACN = 4 were still present at frequencies 15 to 30-fold higher than observed in the original culture of DA81279 grown in MH broth, respectively (Supplementary Data 1).

## Mathematical modeling of amplification copy number distributions

The high frequency of lower ACNs in the distributions following antibiotic exposure was not expected. Since TZP resistance in DA76595 is caused by a β-lactamase, it is possible that cells with higher $bla_{SHV}$ copy numbers degrade the antibiotic in the medium, allowing for the survival and growth of cells with lower copy numbers[34]. Therefore, we hypothesized that indirect resistance (i.e., the ability of a resistant subpopulation to protect a susceptible population) might explain the unexpectedly wide distributions of ACNs observed. To test this hypothesis without introducing any genetic changes that could potentially alter other factors such as fitness, we designed a mathematical model for simulating population ACN distributions. The model included parameters for recombination rates between tandem amplifications, growth rate and fitness cost, resistance level, β-lactamase production, antibiotic concentration and antibiotic degradation for a population containing ACNs between 0 and 9 (See "Methods", Supplementary Figs. 6, 9 and 10, Supplementary Table 5). To achieve relevant parameter values, calculations were based on phenotypic data of the investigated isolate (fitness cost of amplifications, increase in TZP resistance per additional $bla_{SHV}$ copy, estimated recombination rates; Fig. 3b, Supplementary Fig. 6 and Supplementary Data 3) and published literature[35–38]. We then proceeded to generate hypothetical initial population distributions through the model in the absence of antibiotic-mediated selection (Supplementary Fig. 11). This initial population was exposed to antibiotics while either allowing for indirect resistance through β-lactamase production, or without the production of this enzyme. When indirect resistance was provided in the model, the predicted distribution showed wider ranges of copy numbers, presence of bacteria with lower copy numbers, and more evenly distributed frequencies during antibiotic exposure (Supplementary Fig. 12), a result that was robust during sensitivity analysis of other model parameters (Supplementary Fig. 13). Without indirect resistance, the distribution was substantially narrower, centering around higher ACNs (Supplementary Fig. 14).

The mathematical model was then used to test the hypothesis that the slow reversion rate observed during prolonged growth in absence of antibiotics might be due to a low fitness cost associated with the amplification. Experimentally, the fitness cost associated with increased copies of tandem amplifications was too low to be detected (Supplementary Fig. 6) and therefore was estimated to be ≤1.3 × 10$^{-3}$ per copy, a value within previously estimated costs of amplifications[9]. Mathematical simulations using our estimated fitness cost reveals, in agreement with our experimental results, maintenance of the ACN distribution following growth for 32 transfers in absence of antibiotics (Supplementary Fig. 15a), while using higher fitness cost led to a faster reversion of ACNs (Supplementary Fig. 15b).

## Discussion

We report a previously unresolved dynamic distribution of resistance gene tandem amplifications in a HR *E. coli* population and describe an optimized method for direct detection of these genetic events. Our work presents three major new insights in regards to HR and tandem genetic amplifications. First, we show that the phenotype of the HR isolate correlates with the variable copy number of tandem genetic amplifications that are present in subpopulations of cells prior to

exposure to the antibiotic. Second, we describe at single-molecule level the high plasticity of tandem genetic amplifications within bacterial populations, how it is affected by the growth conditions, and its impact on the HR phenotype. Finally, using a mathematical model based on our data, we show that resistance mechanisms and fitness cost of tandem amplifications can affect the distribution of tandem genetic amplifications and therefore the HR phenotype.

Compared to previous studies detecting 4.8 to 17 kbp-long amplified units[22,25,26,28], our approach uses a shorter amplified unit (3.5 kbp), which increases the number of reads sequencing entire arrays of tandem amplifications. By sequencing pDNA rather than whole genome DNA, and by using a unique I-SceI restriction site inserted near the amplified unit, we improved sequencing depth 50-fold over the region of interest. Altogether, our approach allows for the detection of tandem amplifications down to frequencies $10^{-5}$, a never-before-reached resolution of the distribution of tandem genetic amplifications within bacterial populations. The I-SceI restriction site improved the sequencing depth by 22% and allowed for the sequencing of intact plasmids present in the pDNA extraction, which were detected as sequencing reads covering the full length of the plasmid. Thus, we expect the positive impact of I-SceI restriction-digestion on the sequencing depth to increase with the quality of the extracted pDNA, as more plasmids will be extracted in their intact circular form. The amplified unit investigated here, encoding for a SHV-1 β-lactamase surrounded by IS26 elements and natively present as a duplication on a clinical plasmid, represents a genetic arrangement relatively common among Gram-negative pathogens[39].

Growth in presence of increasing concentrations of TZP led to positive selection for increased ACN of an amplified unit encoding the SHV-1 β-lactamase. However, the distribution of ACNs detected was unexpectedly wide. For example, while we measured that ≥8 copies of $bla_{SHV}$ were required for optimal growth in presence of 12 mg L$^{-1}$ TZP, most bacteria in cultures with 12 mg L$^{-1}$ TZP had ACNs <8, and therefore sub-optimal $bla_{SHV}$ copy numbers (Supplementary Fig. 4). Using a mathematical model, we showed that indirect resistance might explain the wide ACN distributions observed here. Indirect resistance, a phenotype where resistant bacteria can protect susceptible bacteria from the effect of antibiotics, has been described for several classes of antibiotics, including β-lactams, tetracyclines, and macrolides[34]. Indirect resistance to β-lactams results from antibiotic inactivation by β-lactamases decreasing the drug concentration in the environment, thus allowing growth of susceptible bacteria. This phenomenon is dependent on the level of expression of the resistance gene[34,40], which will increase with its copy number. Maintenance of populations with very low ACNs through indirect resistance mechanisms might allow faster reversion of ACNs and resistance when increased ACN imposes a substantial fitness cost, by enabling more rapid take-over by cells carrying low ACNs. It could also affect detection of resistance in isolates from patients, as the analyzed colonies might originate from the population of cells with low ACNs that display a low resistance level.

Following the selection for increased ACNs, prolonged growth in absence of TZP did not revert ACNs as fast as seen in previous studies[4,9,19,22]. We showed that the slower reversion observed with cultures initially selected in 12 mg L$^{-1}$ TZP was due to mutants with increased fitness taking over the cultures and slowing down ACN reversion. However, even when such mutants did not dominate the populations, complete ACN reversion following prolonged growth in absence of antibiotics was still not observed. Increased bacterial fitness associated with a decrease in ACNs is expected to be a main driver for reversion of tandem genetic amplifications, and our fitness measurements, based on exponential growth rates and with a sensitivity of ~3%, could not detect a fitness cost associated with these tandem amplifications, even when present at 27 copies. Our mathematical model revealed that, in absence of antibiotics, relatively high ACN

distribution stability was observed when a low fitness cost per amplified unit was applied, but not for high fitness cost values (Supplementary Fig. 15). Thus, we conclude that the slow reversion of increased ACNs that we observed in absence of TZP likely results from a low fitness cost associated with our amplified unit that only encodes 4 genes between the IS26 elements. (Fig. 1a).

The prolonged cultivation of cells in MH broth medium led to the selection of clones carrying additional mutations, most of which had previously been linked to increased fitness in laboratory media. When these fitter clones took over populations, they impacted the ACN distributions by positively selecting populations with distinct ACNs, which could slow down or speed up ACN reversion (and therefore reversion of antibiotic resistance) in absence of antibiotics. Importantly, these results indicate that the distribution and stability of tandem amplifications within populations can be affected by other mutations impacting the fitness of the bacteria. Future studies on the dynamic distribution of ACNs could benefit from using pre-adapted isolates to limit the impact of media-adaptation mutations during prolonged growth.

We acknowledge some limitations of our study. First, one limitation of our detection method is the requirement for very long reads to detect higher ACN and/or to detect arrays of longer amplified units. This can be resolved by increasing sequence coverage and read length (for example, by optimizing pDNA extractions[41] or using a smaller plasmid). Second, as ACN increased, the discrepancy between the average Nanopore-measured copy number of amplified units and the average copy number detected with ddPCR using gDNA extractions increased, and this was due to a lower efficiency of extraction of plasmids with higher copy numbers of amplified units in the pDNA extractions. While we expect that the small discrepancies observed have a limited impact on the results presented here, these might affect the results should higher average ACNs be selected. We propose two hypotheses regarding the origin of the discrepancies observed. Plasmid size increases with the copy number of amplified units (Fig. 2f), reaching up to 199 kbp for the plasmid with the highest ACN detected (ACN = 25, Supplementary Fig. S4). This could decrease the quality and yield of the extracted larger plasmids that are present in bacterial populations and cause the observed discrepancies[42]. This might be minimized by using a smaller plasmid, optimizing pDNA extraction, or using a different E. coli strain[42]. Alternatively, alkaline lysis-based plasmid extraction relies on the sequential denaturation and renaturation of DNA, with the inefficient renaturation of long fragments of chromosomal DNA creating stretches of single-strand DNA that are important for their selective removal during the extraction. Similarly, in bacterial populations carrying a plasmid with varying ACNs, renaturation of plasmids could generate stretches of single-strand DNA when the two hybridizing single DNA strands originated from plasmids with different ACNs. This might lower the efficiency of extracting plasmids carrying high ACNs as these might lead to longer stretches of single-strand DNA following improper renaturation. A third limitation of our study is that while the mathematical model showed similar trends as our data, it sometimes differed regarding the shape of the distributions. For example, modeling led to similar frequencies of ACN = 9 and ACN = 2 following growth in presence of increasing concentrations of TZP (Supplementary Fig. 12). In our experimental data, copy numbers higher than 9 were present (Fig. 1, 3, Supplementary Figs. 4, 5, Supplementary Data 1), while conversion into copy numbers >9 in the model was not possible, leading to possible accumulation of the highest copy number present (ACN = 9) in the simulation (Supplementary Fig. 9).

In conclusion, we revealed a highly dynamic distribution of tandem ACN by using a direct genetic detection method, which allows for the precise determination of copy numbers in a large number of individual cells from a bacterial population. We demonstrated a direct correlation between the distribution of ACNs and the HR phenotype in

the investigated *E. coli* isolate. Moving forward, our approach allows for improved possibilities regarding understanding the parameters that affect the distributions of tandem genetic amplifications and impact ACN-dependent phenomena. Increased knowledge of the genetics behind HR and the factors influencing the distribution of tandem amplifications are important due to the increased risk of antimicrobial treatment failures. It could improve WGS-based prediction of HR[16], the identification of potential high-risk genotypes and, ultimately, the outcome of antimicrobial treatment. More broadly, as tandem gene amplifications are found across all types of organisms and play important roles in both adaptation to new environments and evolution of novel genes, this methodology could also be applied for studies of several other processes. For example, in viruses, gene amplification can increase virulence[43] and in human cancers they can promote tumor formation and resistance to chemotherapeutic agents[44].

## Methods

### Bacterial isolates and genetic construction

DA76595 is a *E. coli* MG1655 carrying the 116 kbp clinical plasmid pDA61218_116 (GenBank ID CP061207.1) originating from the clinical *E. coli* isolate DA61218[19]. The plasmid has a 3.5 kbp-long amplification unit containing a $bla_{SHV-1}$ and flanked by IS26 elements. The plasmid natively carries a duplication of the amplified unit. A 18 nt long recognition site (sequence TAGGGATAACAGGGTAAT) for the restriction enzyme I-SceI was integrated in a neutral locus between genes ICI53_24280 and ICI53_24285, 1659 nucleotides downstream from the amplification through λ-Red recombination. Briefly, an in-house counter-selectable cassette *dhfr-orph11* was integrated 1659 nucleotides downstream from the amplification, and transformats were selected in presence of 10 mg L$^{-1}$ trimethoprim. The counter-selectable marker was then replaced by the I-SceI restriction site using λ-Red recombination and a single-strand long oligonucleotide. Transformants carrying the I-SceI restriction site were selected on non-permissive plates (M9 minimal medium supplemented with 0.2% rhamnose, which induces transcription of the toxin gene *orph11*). The final construct consisting of the I-SceI restriction site isolated from the surrounding DNA by a *lux* and p22 terminator (both remnants of the *dhfr-orph11* cassette) was verified by Sanger sequencing (Eurofins). Primers used for lambda-red and Sanger sequencing were obtained from Merck and are described in Supplementary Table 6.

### Media and antibiotics

Mueller-Hinton broth and agar (Difco) was used for liquid cultures and agar plates. Antibiotics, piperacillin, tazobactam and trimethoprim were purchased from Merck/Sigma-Aldrich. The piperacillin-tazobactam drug combination was used in an 8:1 ratio. Stated concentrations refers to piperacillin concentrations.

### Culturing with and without antibiotic supplementation

Frozen stock (−80 °C, 10% DMSO) of DA76595 was streaked on agar plate and incubated in 37 °C overnight. Isolated colonies were inoculated in 5 mL broth each and incubated for 24 ± 0.5 h in 37 °C, 190 rpm. 450 μL of this culture was then transferred to 450 mL broth without antibiotics and incubated for 24 ± 0.5 h in 37 °C, 190 rpm. Culture was then transferred through increasing concentrations of piperacillin-tazobactam (2 replicates 3, 6 and 12 mg L$^{-1}$, 2 replicates 3, 6 and 9 mg L$^{-1}$) using the same volumes and incubation parameters: 450 μL in 450 mL broth supplemented with antibiotics, followed by 24 ± 0.5 h incubation in 37 °C, 190 rpm, to allow for ±10 generations of growth before next transfer. For all conditions, culture aliquots were saved (−80 °C, 10% glycerol) as well as cell pellets for pDNA and gDNA extraction (stored at −80 °C). For the subsequent culturing without antibiotics, frozen culture aliquots (−80 °C, 10% glycerol) were used for inoculation. The equivalent of 300 μL culture was inoculated in 300 mL broth without antibiotics and incubated for 24 ± 0.5 h in 37 °C, 190 rpm. 300 μL of this culture was then transferred to 300 mL broth. Incubation and transfer procedure was repeated for a total of 32 transfers, (equivalent to ± 320 generations) for all four replicates. If contamination was suspected, culture was restarted using the previous frozen aliquot (−80 °C, 10% glycerol) and same volume parameters. Culture aliquots were saved (−80 °C, 10% glycerol), and cell pellets for pDNA and gDNA extractions (stored at −80 °C) from 10, 20, 40, 80, 160, 240, and 320 generations.

### Population analysis profile (PAP) tests

For phenotypic characterization of DA76595 and comparison towards DA61218, frozen isolate stock (−80 °C, 10% DMSO) was streaked on agar plates and incubated in 37 °C overnight. Broth culture was then started from an isolated colony and incubated in 37 °C, 190 rpm, overnight. This culture was then used to inoculate fresh broth in a 1:1000 ratio, either undilute or after a 1:1000 dilution in phosphate buffered saline (PBS; 8 g L$^{-1}$ NaCl, 0.2 g L$^{-1}$ KCl, 1.44 g L$^{-1}$ Na$_2$HPO$_4$ and 0.24 g L$^{-1}$ KH$_2$PO$_4$), and incubated in 37 °C, 190 rpm, overnight. This final culture was used for plating (see below). For PAP tests of cultures from the Nanopore sequenced populations, cultures were performed as described below (see "Culturing with and without antibiotic supplementation"), and frozen aliquots (−80 °C, 10% glycerol) were used for plating. Before plating, the bacterial cultures were serial diluted in PBS either in 1:10 dilution steps, or an initial 1:20 dilution followed by 1:10 dilution steps. Suitable dilutions were plated on piperacillin-tazobactam supplemented agar plates, either as 3 × 5 μL drops (triplicates) or 100 μL on a full plate using glass beads, as well as agar plates without antibiotics as either 3 × 5 μL or 6 ×5 μL drops. Antibiotic supplemented plates were incubated in 37 °C and plates without antibiotics in 30 °C overnight, whereafter colonies was counted on drop plates while full plates were incubated for an additional 24 h in 37 °C before colonies were counted. Population frequency at each antibiotic concentration was calculated through comparison towards agar plates without antibiotic.

### DNA extractions

DNA was extracted from broth culture cell pellets frozen in −80 °C (see Culturing). Whole genome DNA extraction was done from cell pellets from 1 mL culture using MasterPure™ Complete DNA & RNA Purification Kit (Biosearch Technologies) with RNase treatment according to manufacturer's recommendation with the following modifications: Proteinase K amount was reduced from 1 μL to 0.5 μL, RNase A amount was increased from 1 μL to 1.5 μL and incubation with RNase A was prolonged from 30 min to 90 min, with sample inversion every 15 min. Plasmid DNA extraction was done from ODV800 cell pellets using NucleoBond® Xtra Midi (Macherey-Nagel) in accordance with manufacturer's recommendation. DNA extractions were controlled for impurities using NanoDrop™ spectrophotometer (Thermo Fisher). DNA concentration was measured with Qubit® (Invitrogen) fluorometer, using Qubit® dsDNA Quantitation High Sensitivity or Broad Range kits (Invitrogen). Bead-beating DNA extractions were done as follows: Cell pellets from 360 μL of cultures were resuspended in 1:1 Fast Lysis Buffer for bacterial lysis (Qiagen) and molecular grade water (Sigma). Resuspension was then bead beated using 0.75 μg of 212–300 μM acid washed glass beads (Sigma-Aldrich) together with 250 μL of Phenol:Chloroform:Isoamyl alcohol (25:24:1) (Sigma-Aldrich) in a FastPrep®-24 instrument (MP Biomedicals) for two cycles of 20 s at 6.5 m s$^{-1}$. Vials were then centrifuged at 9600 × g for 3 min at 4 °C. Acqueous phase was transferred to a clean Eppendorf vial. 200 μL of Phenol:Chloroform:Isoamyl alcohol were added, the vials were briefly vortexed and then centrifuged at 12,300 × g at 4 °C for 3 min. The aqueous phase was transferred into a new Eppendorf vial before storing the DNA preparation at −20 °C.

## Plasmid DNA digestion and purification

Before Nanopore sequencing library preparation, plasmid DNA was digested with I-SceI (Thermo Scientific) according to manufacturer's recommendations, followed by purification using AMPure XP (Beckman Coulter) magnetic beads, 1.8 × sample volume, and resuspension in molecular grade water (Sigma). DNA concentration was then measured with Qubit® (Invitrogen) fluorometer, using Qubit® dsDNA Quantitation High Sensitivity kit (Invitrogen). After resuspension an optional, non-necessary PCR control of digestion can be performed using primers flanking the digestion site, comparing the amplification product using undigested and digested DNA of the same sample. For this, we used 0.7 ng DNA template for PCR reaction set up with DreamTaq Green PCR Master Mix 2x (Thermo Scientific), primers (Supplementary Table 6), and molecular grade water according to manufacturer's instructions and following PCR conditions: 95 °C for 1 min, 25 cycles of 95 °C for 30 s. and 55 °C for 30 s. and 72 °C for 1 min, 72 °C for 5 min, 8 °C hold, in a thermocycler block (BioRad). PCR product was then visualized on a 1% agarose (Sigma-Aldrich) gel with GelRed (Biotium) stain and GeneRuler 100 bp (Thermo Scientific) ladder using a Gel Doc (BioRad) UV light system after gel electrophoresis.

## Digital droplet PCR (ddPCR)

Analysis of plasmid copy number (whole genome DNA) and amplification copy number (whole genome DNA and plasmid DNA) was performed using a QX200™ ddPCR system (Bio-Rad). DNA template was pre-digested using SacI-HF restriction enzyme in rCutSmart™ Buffer (New England Biolabs) according to manufacturer's recommendation. Pre-digested template was then used for ddPCR reaction setup with QX200™ ddPCR™ EvaGreen Supermix (Biorad), primers (obtained from Merck; Supplementary Table 6) and molecular grade water (Sigma) according to manufactures recommendations. 20 μL of the reaction mix was used for droplet formation in a DG32 Automated Droplet Generation Cartridge (Biorad) together with Automated Droplet Generation Oil for EvaGreen (Biorad), which generated the drops in a 96 well semi-skirted ddPCR plate (Biorad). Following droplet generation, the ddPCR plate was heat sealed with Pierceable Foil Heat Seal (Biorad). PCR amplification was performed in a thermocycler block (BioRad) under following conditions: 95 °C for 5 min, 40 cycles of 94 °C for 30 s and 56 °C for 1 min, 4 °C for 10 min, 90 °C for 5 min and cooling down to 12 °C. For all steps, a ramp speed of 2 °C s$^{-1}$ was used. QuantaSoft version 1.7.4.0917 (Biorad) or QX Manager Standard edition version 2.2.0 (Biorad) was used for raw data analysis. Plasmid and amplification copy number was calculated through comparison of copies per μL (plasmid normalized towards chromosome and amplification normalized towards plasmid).

## Measuring fitness cost of amplification

Frozen cultures of DA76595 and replicates selected at higher concentrations of TZP were streaked on MHA plate, incubated in 37 °C overnight. Isolated colonies were inoculated into 0.5 mL MH broth and incubated in 37 °C, 190 rpm, overnight. ACN were determined by ddPCR (see "Digital droplet PCR") using template DNA prepared using bead-beating (see "DNA extractions"). Here, we assume that, due to the relative stability of ACNs during prolonged growth in the absence of TZP (Supplementary Figs. 4, 5, 8), they would remain stable during the short growth period in broth. Therefore, the average ACN measured by ddPCR is likely to be close to that of the main populations present in the culture. Fitness (exponential growth rate) cost was determined for each of the cultures using a Bioscreen apparatus (Oy Growth Curves, Finland) as follows: the overnight cutures were diluted 1:1000 into fresh MH broth, and 300 μL was added per well in a Bioscreen plate. Cultures were grown at 37 °C under continuous agitation, which was

paused 5 s before each absorbance measurement, and absorbance was measured every 4 min. Four to six technical repeats were performed per colony tested. BAT was used for analysis of data[45]. The average standard deviation of 4 technical repeats did not allow detection of fitness differences <3%. Using data from the mutants with ACN = 27 copies, we therefore estimated the fitness cost per copy of the amplified unit to be <1 × 10$^{-3}$. Exponential growth rates are typically measured within ≤3 h of growth in MH broth in absence of antibiotics. This short incubation minimizes changes in ACN, which ensures that the fitness measured is that of the main population present in the culture.

## Relation between ACN and resistance level

The same cultures as above were used. The resistance level of the cultures was determined by PAP tests, and the resistance level was determined as the highest concentration of TZP where ≥20% of the plated bacteria were growing. Thus, this resistance level measures the capacity of the main popuation of the cultres to grow in presence of increasing concentrations of TZP. Cultures with different ACNs can have a same measured resistance level as long as the copy number of bla$_{SHV}$ does not allow growth at the next higher-up TZP concentration used in the PAP tests. However, in reality, these cultures carrying different ACNs have real biological resistance levels falling within a continuum of values in-between the measured resistance level and the next higher-up TZP concentration of the PAP test. Thus, among cultures that vary regarding their average ACN but have identical measured resistance levels, those with the lowest ACN will have a real resistance level closest to that measured. Therefore, we used the lowest copy number of tandem amplifications allowing growth on each of the increasing TZP concentrations tested to determine the impact of each additional bla$_{SHV}$ copy on resistance using linear regression analysis performed in Microsoft Excel. We excluded cases where the lowest copy number of amplified units allowing growth at specific TZP concentrations could not be determined accurately, such as for two cultures with contradictory results due to their ACN resulting in a TZP resistance level at the transition between 6 mg L$^{-1}$ and 12 mg L$^{-1}$ (a culture with ACN = 7.25 with a measured TZP resistance of 12 mg L$^{-1}$, and a culture with a ACN = 8.14 and a measured TZP resistance of 6 mg/L; see Supplementary Data 4).

## Estimation of subpopulations frequencies based on ACN distribution data

We used the determined relation between ACN and TZP resistance level (described above) and the ACN distributions detected using Nanopore sequencing to calculate theoretical subpopulations of bacteria that should grow at the different TZP concentrations used for PAP tests. For this, we determined the frequency of bacteria in the populations that carried at least the minimum number of bla$_{SHV}$ copies that are necessary for the bacteria to grow at each of the TZP concentrations used in the PAP tests. These theoretical frequencies of resistant subpopulations determined using the ACN distributions were compared to the frequencies of resistant subpopulations detected using PAP tests.

## Estimations of recombination rates

Calculations were done to estimate potential recombination rates (Supplementary Data 3) between whole tandem units and between IS elements that would lead to similar frequencies of ACN = 0 and ACN = 1 that were observed following Nanopore sequencing of 5 replicates of DA76595 grown in MH broth. The isolated colonies used for starting each of the 5 cultures originated from single bacterium carrying a duplication (ACN = 2). From this initial bacterium, we estimated ±45 generations of growth until Nanopore sequencing was performed: ±25 generations of growth in the colony, ±10 generations of growth for the

initial overnight, and 10 generations of growth to get the culture used for Nanopore sequencing. No important bottlenecks appeared during the cultures as inoculates (colonies and 450 μL of cultures) contained >$10^8$ bacteria. We considered that frequencies of ACN ≠ 2 appearing in the cultures during growth mostly resulted from formation and reversion rates, with a limited impact of fitness differences of bacteria harboring different ACNs since (i) we could not measure any fitness difference between bacteria with different ACNs (Supplementary Fig. 6), and (ii) ACN distributions were relatively stable over many growth generations in MH broth in absence of TZP (Supplementary Figs. 4, 5, 8), a phenomenon that is dependent on bacteria with lower ACNs having no or only marginal fitness increases. However, we acknowledge that not taking into account the fitness cost of the tandem amplifications only allows for a rough estimation of the recombination rates. The frequency of ACN = 1 after 45 generations was used to estimate the rate of conversion from ACN ≥ 2 to ACN = 1 (recombination between whole amplified units), and the frequency of ACN = 0 in the population after ±45 generations was used to estimate the rate of conversion from any ACN ≥ 1 to ACN = 0 (recombination between IS26 elements present on each end of the amplified unit).

## Short read sequencing

Whole genome gDNA was short read sequenced through DNA Nano-Ball Sequencing (DNBSEQ) without PCR amplification and was performed by BGI. Sequencing data was analyzed for read coverage and presence of mutations using CLC Genomics Workbench (Quiagen). Plasmid and amplification copy number was calculated through coverage comparison (plasmid normalized towards chromosome and amplification normalized towards plasmid).

## Nanopore long read sequencing

Sequencing of I-SceI-digested pDNA was performed using Oxford Nanopore Technologies R10 MinIon flow cells and V14 library kits on a Mk1c device. For single sample sequencing, libraries were prepared using the Ligation sequencing kit V14, with 1.5 μg of DNA as starting input. Library was prepared according to manufacturer's recommendation with the following modifications: During Adapter ligation and clean-up, (i) 42 μL instead of 40 μL AMPure XP beads was used, (ii) Short Fragment Buffer was used, and (iii) incubation in Elution Buffer was done in 37 °C for 15 min. For multiplexed sequencing, 12 sample libraries were prepared using the Native Barcoding Kit 96 V14 kit, with 400 ng digested plasmid DNA per sample as starting input. Library was prepared according to manufacturer's recommendations with the following modifications: During Native barcode ligation and clean up, (i) the volume used from the foregoing DNA repair and end prep was scaled up x8, together with same scaling of the reagents used during the Native barcoding ligation step, resulting in a total volume and DNA amount equivalent to 96 samples in the library, (ii) 0.42x sample volume instead of 0.40x of AMPure XP Beads was used and (iii) Molecular grade water resuspension of magnetic bead pellet was incubated in 37 °C for 15 min. During Adapter ligation and clean up: (i) 0.42x sample volume instead of 0.40x of AMPure XP beads was used, (ii) Short Fragment Buffer was used and (iii) incubation in Elution Buffer was done in 37 °C for 15 min. Sequencing of undigested plasmid DNA was performed using R9 MinIon flow cell and v9 Ligation Sequencing library kit, single sample, 1 μg purified plasmid DNA as starting input, on a Mk1c device. Library was prepared according to manufacturer's recommendation.

## Bioinformatic pipeline for analysis of Nanopore long read sequencing

Our approach consisted of several BLAST iterations and filtering steps. The entire analysis pipeline was written using Snakemake v8.23.1 workflow management system, Python v3.12.10 and R v4.4.1 scripts[46].

To ensure the analysis reproducibility we provide both conda environments (https://docs.anaconda.com/working-with-conda/environments/) and Docker/Apptainer containers. All code used for this analysis along with configuration files, test data set, and software versions are available on the projects GitHub repository https://github.com/andrewgull/tga_detection.

As input, the pipeline takes the following setup (all coordinates relative to CP061207.1):

1. Samples: A list of samples to analyse.
2. Minimum Nanopore Read Length: The minimum read length for nanopore sequencing is set to 1820 nucleotides, calculated as the sum of two flanking regions (FR) and an insertion sequence (IS) plus an additional 10%.
3. Flanking Regions (FR) Locations:
   - Flanking Region 1 (FR1): Located between positions 2785 and 3285, with a length of 500 nucleotides.
   - Flanking Region 2 (FR2): Located between positions 114451 and 114951, also with a length of 500 nucleotides.
4. IS Location: The repetitive unit is situated between positions 1955 and 2774.
5. $bla_{SHV}$ Gene Location: The $bla_{SHV}$ gene spans from position 1 to 861.
6. BLAST Settings for Flanking Regions (FR) and $bla_{SHV}$:
   - The output format is set to 6.
   - The number of alignments for FR is capped at 10,000,000.
   - The number of alignments for $bla_{SHV}$ is capped at 10,000,000.
7. BLAST Filtering Criteria:
   - Minimum length for FR alignments is 500 nucleotides.
   - Minimum identity for alignments is 75%.
   - Maximum E-value for alignments is 0.00001.
   - Minimum length for IS alignments is 700 nucleotides.
   - Maximum distance allowed between alignments is 30 nucleotides.
   - Base length for calculations is 1820 nucleotides.
   - Maximum copy number (CN) is set to 60, with an increment value of 3450. This value should be increased if the frequency table contains NAs.
8. Bedtools Merge Parameters[47]: The distance for merging intervals is set to 500 nucleotides.

Specifically, the pipeline consists of the following steps (see Supplementary Fig. 2):

1. Reads shorter than 1820 nt are discarded because they possibly could not contain both flanking regions and the insertion sequence (FR1 + IS + FR2) – and therefore are not relevant to the study.
2. Both flanking regions are BLASTed against the given set of reads, and the BLAST tables (output format 6) are collected.
3. The IS is BLASTed against the given set of reads, and the BLAST table are collected.
4. Using these tables with BLAST produced hits, we keep only the reads containing both FR1 and FR2 in the same orientation. The maximal distance between FR1 and IS was set to 30 nt, minimal length of IS hit was set to 700 nt, minimal length of FR1 and FR2 hit was set to 500 nt, minimal identity of BLAST hits was set to 75%, maximal BLAST E-value was set to $10^{-5}$.
5. Reads with the distance between the FR1 (excluding the FR1) and the end of the read (in the direction of the IS) shorter than 1320 nt are discarded as they cannot contain a correctly detected FR2.
6. Reads containing FR2 in correct orientation relative to both FR1 and IS are kept for further analysis.
7. Number of reads possibly containing every $bla_{SHV}$ copy number variant was inferred from the read lengths to be later used in the calculation of detection limit. Theoretical frequency of reads

containing each copy-number variant was calculated using the data.

8. Discard reads with more than one FR1 or FR2 (artifact hybrid reads formed during sequencing library preparation).

9. Nucleotide sequence of $bla_{SHV}$ gene is BLASTed against the remaining reads; $bla_{SHV}$ hits with $E$-value greater than $10^{-5}$ were discarded.

10. Reads containing $bla_{SHV}$ hits outside the FR1 - FR2 region were discarded from the analysis.

11. The $bla_{SHV}$ hits were merged with the maximum distance between hits to be merged was set to 500 nt and the number of $bla_{SHV}$ genes detected was counted.

12. Expected number of $bla_{SHV}$ genes according to the distance between FR1 and FR2 were calculated using Eq. (1)

$$Nbla\_\exp = ((DFR - 4299)/3450) + 1, \qquad (1)$$

where DFR is the distance between the flanking regions. Reads where the expected number of $bla_{SHV}$ genes and the measured number of $bla_{SHV}$ gene (by BLAST analysis) differed by more than one $bla_{SHV}$ were discarded.

13. From these data, observed count and frequency of reads containing each copy-number variant was calculated.

14. Using the theoretical data from step 7, the observed counts and frequences (from step 12) were corrected by dividing the observed counts by theoretical frequency.

15. The limits of copy-number detection were calculated as inverse of the number of reads that can theoretically contain each copy-number variant (from step 7).

Additionally, inside the pipeline the following software packages were used: for BLAST search we used BLAST+ package v2.12.0[48,49]; for filtering and handling Nanopore reads we used Filtlong v0.2.1 (https://github.com/rrwick/Filtlong/tree/v0.2.1) and Seqkit v2.0.0[50] and R package Biostrings v2.70.1[51].

Our bioinformatic pipeline includes several innovative steps for optimal ACN detection. First, Nanopore over-sequences short reads to the detriment of long reads, likely due to DNA fragmentation during extraction and library preparation, and faster sequencing of shorter reads. This can lead to the over-detection of low ACNs and the under-detection of high ACNs that rely on the sequencing of longer reads for their detection. Steps 7 and 14 of our pipeline take that bias into consideration by adjusting the ACN frequencies detected to the Nanopore sequencing read length distribution observed. Second, the ligation-based library kit used might ligate together different DNA fragments present in the pDNA extraction. This could create aberrant reads that might affect our data, especially for ACNs detected at very low frequencies. Steps 8, 10, and 12 of our pipeline remove those rare reads, ensuring that our analysis focuses on tandem amplifications only. Here, we did not analyse these rare aberrant reads to determine if they might result from true recombination events involving the region of pDA61218_116 encoding $bla_{SHV}$.

## Mathematical modeling and computer simulations
Our mathematical model of the pharmaco- and population dynamics of recombination-based amplification mediated HR is based on two employed mathemtical models. We assume a Hill function for the relationship between the concentration of the antibiotic[38], the concentration of a single limiting resource[36], and the rates of growth and death of the bacteria[35] for modeling the pharmacodynamics of antibiotic treatment[37].

**Pharmacodynamics.** In accord with the Hill function, the net rate of growth or death of a bacterial population given to a given antibiotic concentration is given by Eq. (2).

$$\Pi_i(A, r) = v_{MAXi} - \left[ \frac{(v_{MAXi} - v_{MINi}) \cdot \left(\frac{A}{MIC_i}\right)^{ki}}{\left(\frac{A}{MIC_i}\right)^{ki} - \left(\frac{v_{MINi}}{v_{MAXi}}\right)} \right] \cdot \psi(r) \qquad (2)$$

where $A$ in mg L$^{-1}$ is the antibiotic concentration and $r$ in mg L$^{-1}$ is the concentration of the resource which limits the growth of the population. $v_{MAXi}$ is the maximum growth rate in cells per hour of the bacteria of state $i$, where $v_{MAXi} > 0$. $v_{MINi}$ is the minimum growth rate per cell per hour, which is the maximum death rate when exposed to the antibiotic, where $v_{MINi} < 0$. $MIC_i$ is the minimum inhibitory concentration of the antibiotic for the bacteria of state $i$ in mg L$^{-1}$. $\kappa i$ is the Hill coefficient for bacteria of state $i$. The greater the value of $\kappa i$, the more acute the function. The function, $\psi(r) = \frac{r}{(r+k)}$, is the rate of growth in the absence of the antibiotic, where $k$ is the resource concentration in mg L$^{-1}$ when the growth rate is half of its maximum value. $\psi(r)$ measures the physiological state of the bacteria; as the resource concentration declines the cells grow slower[36].

**Diagrams of the recombination-based resistance model.** The model of recombination-based resistance used here is depicted in Supplementary Fig. 9. States with increasing resistance levels are generated by a transition from a less resistant state to a more resistant state. The states are generated by recombination between sister chromatids[52]. Each state produces a different amount of β-lactamase in a manner proportional to the gene copy-number. One state does not produce any β-lactamase and nine other states produce increasingly more β-lactamase and have an increasingly higher resistance levels. Transition occurs between each of these states in a sequential manner in both directions such that the rate of transition is proportional to the number of ways a recombination can occur which yields that number of gene copies.

**The recombination-based resistance model.** In this model (Supplementary Fig. 9), the bacteria transition between ten different states: $R_0$, $R_1$,..., $R_9$, which are the designations and densities in cells per mL of bacteria of these different states. Cells of the $R_i$ state transition to states $R_x$ and $R_y$ at rates $\mu_{ix}$ and $\mu_{iy}$ per cell per hour, respectively. Transitions also occur in the reverse direction to $R_i$ from states $R_x$ and $R_y$ at rates $\mu_{xi}$ and $\mu_{yi}$ per cell per hour, respectively. The ability to transition between states is limited by the copy number of the gene present on each chromatid. The rate of transition between states is proportional to the number of ways recombination events can generate daughter cells with that specific copy number. A diagram of the mechanism of transition is presented in Supplementary Fig. 10A, B. An explanation of the proportionality of the transition rate is presented in Supplementary Fig. 10C. A matrix of all the possible transitions and their rates is presented in Supplementary Fig. 10D.

We assume the amplified unit is surrounded by identical repeats that can be involved in recombination events leading to the transition between states. The rates of transition between states are solely dependent on the amount of DNA available for recombination[53]. Thus, recombination between whole amplified units is more frequent than the recombination between shorter repeated sequences surrounding each amplified unit. Consequently, the rates of transition to the $R_0$ state and the rates of doubling or halving the copy number of the amplified units (e.g., transition from $R_1$ to $R_2$ or $R_4$ to $R_2$) is likely extremely low since, (i) they result from specific recombination events between short repeated sequences bordering the amplified unit compared to all other transitions between other states, which result from the recombination of larger amplified units, and (ii) the $R_1$ state is very stable in the genome of bacteria in the absence of selection pressure for the presence of the amplified unit. We also assume this mechanism is responsible for all transitions, thus the rates will be

directly proportional to the number of ways recombination can occur to generate that state. The rates of transition were estimated according to observed data (Supplementary Data 3).

We simulate these transitions with a Monte Carlo process. A random number $x$ ($0 \leq x \leq 1$) from a rectangular distribution is generated. If $x$ is less than the product of the number of cells in the generating state ($R_i$, the density times the volume of the vessel, Vol), the transition rate ($\mu$) and the step size ($dt$) of the Euler method employed for solving differential equation, for example if $x < R_1 * \mu_{SR1} * dt * \text{Vol}$, then MR12 cells are added to the $R_2$ population and removed from the $R_1$ population where $MR12 = 1/(dt * \text{Vol})$. With these definitions, assumptions, and the parameters defined and presented in Supplementary Table 5, the rates of change in the densities of the different populations will be given by Eqs. (3–13):

$$\frac{dr}{dt} = -e \cdot \psi(r) \cdot \left( \begin{array}{l} v_{R0} \cdot R_0 + v_{R1} \cdot R_1 + v_{R2} \cdot R_2 + v_{R3} \cdot R_3 + v_{R4} \cdot R_4 + \\ v_{R5} \cdot R_5 + v_{R6} \cdot R_6 + v_{R7} \cdot R_7 + v_{R8} \cdot R_8 + v_{R9} \cdot R_9 \end{array} \right) \quad (3)$$

$$\frac{dR_0}{dt} = R_0 \cdot \Pi_{R_0}(r, A) + (MR10 + MR20 + MR30 + MR40 \\ + MR50 + MR60 + MR70 + MR80 + MR90) \quad (4)$$

$$\frac{dR_1}{dt} = R_1 \cdot \Pi_{R_1}(r, A) + (MR21 + MR31 + MR41 + MR51 + MR61 \\ + MR71 + MR81 + MR91) - (MR12 + MR10) \quad (5)$$

$$\frac{dR_2}{dt} = R_2 \cdot \Pi_{R_2}(r, A) + (MR12 + MR32 + MR42 + MR52 + MR62 \\ + MR72 + MR82 + MR92) - (MR20 + MR21 + MR23) \quad (6)$$

$$\frac{dR_3}{dt} = R_3 \cdot \Pi_{R_3}(r, A) + (MR23 + MR43 + MR53 + MR63 + MR73 \\ + MR83 + MR93) - (MR30 + MR31 + MR32 + MR34 + MR35) \quad (7)$$

$$\frac{dR_4}{dt} = R_4 \cdot \Pi_{R_4}(r, A) + (MR24 + MR34 + MR54 + MR64 + MR74 \\ + MR84 + MR94) - (MR40 + MR41 + MR42 \\ + MR43 + M45 + MR46 + MR47) \quad (8)$$

$$\frac{dR_5}{dt} = R_5 \cdot \Pi_{R_5}(r, A) + (MR35 + MR45 + MR65 + MR75 + MR85 + MR95) \\ - (MR50 + MR51 + MR52 + MR53 + M54 + MR56 + MR57 + MR58 + MR59) \quad (9)$$

$$\frac{dR_6}{dt} = R_6 \cdot \Pi_{R_6}(r, A) + (MR36 + MR46 + MR56 + MR76 + MR86 + MR96) - (MR60 \\ + MR61 + MR62 + MR63 + M64 + MR65 + MR67 + MR68 + MR69) \quad (10)$$

$$\frac{dR_7}{dt} = R_7 \cdot \Pi_{R_7}(r, A) + (MR47 + MR57 + MR67 + MR87 + MR97) \\ - (MR70 + MR71 + MR72 + MR73 + M74 \\ + MR75 + MR76 + MR78 + MR79) \quad (11)$$

$$\frac{dR_8}{dt} = R_8 \cdot \Pi_{R_8}(r, A) + (MR48 + MR58 + MR68 + MR78 + MR98) \\ - (MR80 + MR81 + MR82 + MR83 + M84 \\ + MR85 + MR86 + MR87 + MR89) \quad (12)$$

$$\frac{dR_9}{dt} = R_9 \cdot \Pi_{R_9}(r, A) + (MR59 + MR69 + MR79 + MR89) - (MR90 + MR91 \\ + MR92 + MR93 + M94 + MR95 + MR96 + MR97 + MR98) \quad (13)$$

**Simulated population dynamics.** We first determine the distribution of $R_0$, $R_1$, $R_2$,…, and $R_9$ obtained when a single cell of $R_2$ is grown up to a nutrient-limited density. As these simulations are stochastic, they were each ran 10 times. 1/100th of the average of these 10 simulations will be used as the starting population in all subsequent simulations, representing the initial distribution that would occur when starting an experiment with 1/100th of a stationary-phase, overnight culture. We then simulate the change in density over 3 days when transferred each day in the presence of 3, 6, and 9 mg L$^{-1}$ of a bactericidal antibiotic, progressively. To demonstrate that the relative equal frequency distribution is a product of the mechanism of HR (β-lactamase production) rather than a product of the mechanism of recombination, we perform simulations both with and without β-lactamase production. In addition, we perform a sensitivity analysis by: assuming the change in β-lactamase production is logarithmic with copy number rather than linear; allowing each increase in copy number to convey a 10% fitness cost; and, allowing for a ten-fold higher or lower baseline mutation rate. We also evaluate the stability of the distribution present at 72 h under increasing antibiotic treatment by serially transferring 1/100 without antibiotics for 32 days, for both the estimated and a high fitness cost (each increase in copy number conveys a 5% fitness cost).

**Numerical solutions (simulations).** For our numerical analysis of the coupled, ordered differential Equations presented (Eqs. (3–13)) we used Berkeley Madonna with the parameters presented in Supplementary Table 1[54]. Copies of the Berkeley Madonna programs used for these simulations are available at eclf.net.

### Reporting summary
Further information on research design is available in the Nature Portfolio Reporting Summary linked to this article.

## Data availability
The data generated in this study are provided in the Source Data and the Supplementary Data files. The raw sequences generated in this study have been deposited in the sequence read archive (SRA) database at NCBI (Bioproject PRJNA1299340; https://www.ncbi.nlm.nih.gov/sra/PRJNA1299340). Source data are provided with this paper.

## Code availability
All code used for this analysis along with configuration files and software versions are available on the projects GitHub repository https://github.com/andrewgull/tga_detection.

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

## Acknowledgements

The authors would like to thank Dr. Matthew Lukenge for preliminary work on DA61218 carrying the pDA61218_116 plasmid, Dr. Jennifer Jagdmann for conjugating pDA61218_116 into MG1655 and Pr. Bruce Levin for his help in converting the mathematical model to serial-transfer and for reviewing the mathematical model methods and results. This work was funded by grants to DIA from the Wallenberg Foundation (2018.0168), the Swedish Research Council (2021-02091), and the NIH (1U19AI158080-01).

## Author contributions
H.N. contributed the original idea. S.J., H.N., and D.I.A. designed the study. S.J. performed the majority of the experiments. S.J. and H.N. analyzed the data. A.G., S.J., and H.N. designed the bioinformatics pipeline. A.G. wrote the bioinformatics pipeline and performed the bioinformatics analysis. B.B., S.J., and H.N. designed the mathematical model. B.B. wrote the mathematical model and performed the calculations. S.J. wrote the initial manuscript draft. All authors revised the manuscript. D.I.A. acquired funding.

## Funding

## Competing interests
The authors declare no competing interests.
