## [Transparent Peer Review file · Nature Communications]

The dynamic distribution of genetic tandem amplifications in a heteroresistant *Escherichia coli* population revealed by ultra-deep long read sequencing

Corresponding Author: Dr Hervé Nicoloff

Version 0:

Reviewer comments:

Reviewer #1

(Remarks to the Author)

This is an interesting manuscript aiming to study the dynamics of population amplicon copy numbers at a single cell level using long-read sequencing. The methodology is innovative, but some of the method details and interpretations are unclear. The following are my major concerns:

1. The authors measure amplification dynamics in a specific system where the amplified gene encodes a beta-lactamase and is present on a likely single-copy plasmid in *E. coli*. The generality of their findings in other systems has not been tested. Thus, these details (the species name, the fact that the amplified gene is on a plasmid, and that it encodes a beta-lactamase) should be included in the abstract to make it clear what system is being studied.

2. On lines 177-188, the authors first suggest that the discrepancy between the short read population (WGS and ddPCR) vs long-read Nanopore data was due to differences in plasmid extraction between pDNA and gDNA extraction methods. But they then conclude that their detection based on gDNA did not introduce any bias in the average copy number detected. It is thus not clear where the discrepancy arose.

Along the same lines, is there any way to determine the efficiency or consistency of extraction of plasmid DNA by the pDNA vs gDNA extraction methods? Does this cause any bias in the data?

3. For the analysis of copy number vs TZP-resistance, it is not clear why the authors calculated the increased TZP-resistance level based on the least number of copies that allowed growth at a specific TZP concentration – that suggests a simple direct correlation between the two variables. However, the graph shown in Fig. 3B demonstrates that the relationship between copy number and level of TZP resistance is likely not a straightforward one and depends on other factors. It would be better if linear regression was performed on the entire dataset, and not just the lowest copy number, to account for other factors as well.

Also, to figure out the relationship between the copy number and resistance, the ACN of individual colonies was measured by ddPCR. However, it is likely that there is already heterogeneity in ACN within a colony, given the number of divisions that are required to go from a single cell to a colony. Did the authors measure the ACN of a colony using long-read sequencing to test for population heterogeneity?

4. In the calculation of recombination rates, the authors estimate that conversion from ACN ≥ 2 to ACN=1 occurred due to recombination between whole amplified units. However, this could also occur due to preferential selection against cells with higher copy numbers. How do the authors rule this out?

This is also true for their data that shows that selection for fitter variants led to increased prevalence of populations with distinct ACNs (Fig. S8). If the tandem copies in a cell were recombining to alter the ACN, any increased prevalence of a specific ACN (containing the fitter variants) would be quickly lost as the ACN changed within the population.

5. Long read sequencing has a bias against longer reads – how do the authors normalize for such a bias (which would

predominantly affect reads corresponding to higher ACNs)?

Further, in Figures 3, S4, and S5, it appears that most of the high ACNs detected are at the limit of detection – and the Supplementary Dataset shows that in several cases there was only one read that corresponded to the highest copy number. Perhaps if the bias against longer reads can be quantified, the data could be normalized to account for such a bias, but as the data stands now, it is difficult to interpret data with single reads for a lot of data-points.

6. During the fitness measurement, strains were grown without antibiotic and growth was measured. However, the authors only measured the ACN by ddPCR before the growth assay. It is possible that the ACN decreased to similar levels across all strains due to the absence of selection, which then led to similar fitness across the various colonies. Can the authors also do ddPCR / long read sequencing after the growth to show that the ACN did not change during the experiment? Further, the fitness measurements were only performed as technical replicates, with no biological replicates.

Minor concerns:

1. In Fig. S1, only one replicate is shown. Given that the plasmid is introduced into a different strain, and has been genetically modified, it is interesting that it has a similar effect in both strains. It would thus be useful to show 3 independent replicates each.

2. On line 111, the authors say that “sequencing of pDNA rather than of whole genomic DNA (gDNA) increases the sequencing depth by a theoretical 41-fold” – is this just due to the relative sizes of the genome and plasmid? Some details should be provided.

3. In Table S2, what is “sample size”?

4. For Figure 3C, the authors should also describe how they estimated the frequencies of mutants that should grow at the different TZP concentrations.

5. Figure legends for Figures S11-15 should clarify that these data are from a model and are not experimental data.

(Remarks on code availability)

Reviewer #2

(Remarks to the Author)

Summary

This is an interesting and clearly written manuscript. While other groups have used long-read nanopore sequencing to report variable cell-to-cell amplification copy numbers in heteroresistant populations, this study achieves an unprecedented level of resolution on the distribution of tandem genetic amplifications within bacterial populations. The authors accomplished this using a novel methodology. Although their approach is limited to amplifications on plasmids, this is, to the best of my knowledge, the first time these copy number variations have been observed at this scale. Notably, their simple yet clever innovation of introducing an I-SceI restriction site adjacent to the amplification site improved read coverage. As is typical of this group, their literature review is thorough and relevant to their findings.

Major comments

As noted above, the novel approach presented requires that amplifications are plasmid-borne. These authors report the dynamics of amplification copy number distribution in the presence and absence of an antibiotic; however, understanding the relevance of their findings is somewhat dependent on establishing the clinical prevalence of plasmid-borne amplifications. The authors state that “amplification-mediated HR in *E. coli* is frequently plasmid-associated” and that plasmid-borne amplifications are “relatively common among Gram-negative pathogens” but do not provide quantitative details. To enhance the manuscript, it would be beneficial to either:

Provide additional genomic analysis of clinical isolates to quantify the prevalence of this mechanism.

Reference specific, previously published studies that directly quantify its occurrence in clinical settings.

A key claim throughout the manuscript is the ability to “resolve single cell amplification copy numbers.” This assertion hinges on the assumption that the plasmid containing the amplification unit is present at a stable single copy per cell within the population. It is important for the authors to provide supporting evidence or a robust justification for this assumption. Without this, the data more accurately represent single-molecule (or single-plasmid) resolution rather than true single-cell copy number.

In Figure S2 and Methods, the authors describe discarding artifact hybrid reads formed during sequencing library preparation, specifically those with multiple FR1 or FR2 sequences, or blaSHV hits outside the FR1-FR2 region. Given the high sequencing depths, it's possible to capture other rare and complex structural variants alongside the tandem amplifications. The manuscript does not clearly explain how the authors distinguish between these genuine structural variations and the discarded artifact hybrid reads. Could these artifacts also affect reads containing the tandem amplification flanked by FR1 and FR2? Further clarification on the criteria used to make this distinction is needed.

Minor comments

Lines 97-100. Text is at times redundant. For example, is this sentence really needed: The ability to detect rare tandem amplification events using Nanopore sequencing is dependent on the sequencing depth over the area of interest and on sequencing reads being long enough to cover the whole array of amplified units. This concept was explained in detail in the Introduction.

Figure 1A and 1B would greatly benefit from the addition of scale bars. Since the method presented depends on generating reads long enough to span the amplified arrays, the reader should be able to easily understand the scale of the amplified unit and its relationship to the engineered plasmid.

Line 123 Typo hypotesized -> hypothesized

(Remarks on code availability)

I did not attempt to install or run the code but it appears to be well-documented and organized such that it could be a usable resource for the community.

Reviewer #5

(Remarks to the Author)

This study provides valuable insights into the molecular mechanisms of heteroresistance, a clinically relevant but still poorly understood phenomenon. The authors focus on the dynamics of tandem gene amplification of the SHV-1 β -lactamase in *Escherichia coli*, investigating its dynamics in a single-copy plasmid originally found in a clinical isolate. To facilitate the analysis, the plasmid was transferred into the laboratory strain *E. coli* MG1655. Using a plasmid version that was then engineered with a unique restriction site, they established a novel robust protocol that enabled high-coverage sequencing by Nanopore Long-read sequencing of the genomic region prone to tandem amplification. Following plasmid extraction and enzymatic digestion, the authors applied sequencing-by-ligation and deep nanopore long-read sequencing. Digital droplet PCR and short-read sequencing were used to compare allele copy number estimates. This study describes interesting aspects of tandem amplification dynamics and highlights the power of nanopore long-read sequencing for resolving this type of complex genomic rearrangements. Importantly, the single-copy nature of the plasmid combined with long-read technology allows single-cell resolution of amplification events.

The main limitation of this otherwise elegant protocol is its reliance on the presence of a unique restriction site in close proximity to the resistance gene undergoing tandem amplification. This requirement may impair the generalizability of the approach to other plasmids and clinical isolates where such sites are absent or not easily engineered. It would be valuable if the authors could comment on how broadly applicable they envision this strategy to be, and whether alternative approaches might overcome this limitation.

Specific comments:

Result section:

- Please ensure that all sequencing data are deposited in a public repository and provide the corresponding accession number
- L102: Was the plasmid also single copy in the original clinical isolate?
- L106: Since plasmids were extracted prior to digestion with I-SceI, what was the rationale for selecting an enzyme that does not cut the rest of the genome?
- L117: Given that sequencing depth improved by only 22% with the digestion step, have the authors attempted the method without digestion? This could make the approach more broadly applicable to clinical isolates.
- L163: Including the number of passages alongside the number of generations (at least once in the Results section) would help the reader contextualize the data
- L243: The authors suggest that indirect resistance might explain the wide distribution observed. Could they hypothesize further on the origin of this indirect resistance? For example, were they able to detect free SHV-1 enzyme degrading the antibiotic in the culture medium?
- L180–187: The section addressing ACNs discrepancies between Nanopore sequencing of plasmid DNA and short-read WGS/ddPCR on genomic DNA is unclear. Methods of lysis and DNA extraction are compared without further details and mixed within the same sentences. Please clarify. Are these discrepancies potentially linked to plasmid length, where larger plasmids may be sub-optimally recovered using plasmid extraction kits?
- L211: Did the authors sequence the wild-type strain after passaging in antibiotic-free media as a control?

Method section:

- L645: The description of mathematical modeling and computer simulation methods could be moved to the Supplementary Information, as most readers (including myself) are unlikely to be able to evaluate these details.

Figures:

- A figure displaying representative reads (e.g., using IGV, Integrative Genomics Viewer) showing tandem amplification would greatly help readers visualize the amplification on single reads.
- Figure 1B: The locations of CR5 and CR4 loci are difficult to discern; could these also be indicated in Figure 1A for improved clarity?
- Figure 1D: Please clarify the figure legend, as the description is currently much clearer in the text.

(Remarks on code availability)

I have not reviewed the code, as this falls outside my area of expertise.

Version 1:

Reviewer comments:

Reviewer #1

(Remarks to the Author)

This is a revised version of a manuscript examining dynamics of gene amplification leading to antibiotic resistance, and amplified copy numbers, in the presence and absence of antibiotics. The authors have addressed almost all of my previous concerns. I have one follow-up query below, and noticed a couple of typos:

1. I had previously raised a concern about the analysis of copy number vs TZP-resistance, querying why the authors calculated the increased TZP-resistance level based on the least number of copies that allowed growth at a specific TZP concentration. The authors' response was that while ACN is a continuous variable, the measured resistance level follows a discrete distribution. Thus, the real biological resistance of the populations with different ACNs falls somewhere in the continuum between the measured resistance level and the next higher-up resistance level.

While this explanation theoretically makes sense, it doesn't perfectly fit their data. E.g. they conclude that an ACN of ~7 should result in resistance level of 12 mg/L. However, they have an isolate with higher ACN numbers whose resistance level is only 6 mg/L (Figure 3b). Does this imply that other factors also alter resistance levels? This should be mentioned.

2. Line 99: typo in 'previously'

3. Line 569: typo in 'vary'

(Remarks on code availability)

Reviewer #2

(Remarks to the Author)

Overall, it is my view that the the revised manuscript adequately addresses the concerns raised by the reviewers.

With respect to the comment below, I think it is more accurate to describe detection of tandem amplifications at the single molecule, rather than single cell, level. ddPCR and WGS report the average plasmid copy number in the population and I do not think you can strictly speaking rule out changes in cell-to-cell plasmid number on the basis on these data.

Reviewer2: A key claim throughout the manuscript is the ability to "resolve single cell amplification copy numbers." This assertion hinges on the assumption that the plasmid containing the amplification unit is present at a stable single copy per cell within the population. It is important for the authors to provide supporting evidence or a robust justification for this assumption. Without this, the data more accurately represent single-molecule (or single-plasmid) resolution rather than true single-cell copy number.

Authors: Our manuscript includes data about the plasmid copy number, which we determined throughout the experiments. We have now updated the text to clarify our claim regarding detection of tandem amplifications at single cell level (L179-182).

(Remarks on code availability)

I have successfully run the code and can confirm that it works as specified.

I encountered a minor issue with Snakemake; I was unable to get it to work as described in the GitHub comment: "NB: Given that you have Snakemake installed, all other dependencies will be installed and deployed automatically when you run the pipeline (see below)." As a result, I had to install the dependencies manually. This is not a critical issue, as the code ultimately ran correctly.

I also noted a "\\n" character at the end of the FASTA file in the test data, which initially caused the run to stall. This should be an easy fix for the authors.

Reviewer #5

(Remarks to the Author)

The authors have provided the additional information requested, and my previous concerns have been adequately addressed. I have no further major comments.

(Remarks on code availability)

The dynamic distribution of genetic tandem amplifications in a heteroresistant *Escherichia coli* population revealed by ultra-deep long read sequencing

Sofia Jonsson, Andrei Guliaev, Brandon A. Berryhill, Dan I. Andersson, Hervé Nicoloff

ANSWERS TO REVIEWERS COMMENTS

Reviewer #1 (Remarks to the Author):

This is an interesting manuscript aiming to study the dynamics of population amplicon copy numbers at a single cell level using long-read sequencing. The methodology is innovative, but some of the method details and interpretations are unclear. The following are my major concerns:

We thank the reviewer for the positive words.

1. The authors measure amplification dynamics in a specific system where the amplified gene encodes a beta-lactamase and is present on a likely single-copy plasmid in *E. coli*. The generality of their findings in other systems has not been tested. Thus, these details (the species name, the fact that the amplified gene is on a plasmid, and that it encodes a beta-lactamase) should be included in the abstract to make it clear what system is being studied.

The information has been added to the abstract (L22, 24-27).

2. On lines 177-188, the authors first suggest that the discrepancy between the short read population (WGS and ddPCR) vs long-read Nanopore data was due to differences in plasmid extraction between pDNA and gDNA extraction methods. But they then conclude that their detection based on gDNA did not introduce any bias in the average copy number detected. It is thus not clear where the discrepancy arose.

We apologize for the confusion. Our results indicate that the discrepancy arose during the pDNA extraction itself. This part in the text has now been clarified (L187-201).

Along the same lines, is there any way to determine the efficiency or consistency of extraction of plasmid DNA by the pDNA vs gDNA extraction methods? Does this cause any bias in the data?

When analysing the data from Table S1, there was a strong positive correlation between the average ACN detected in the gDNA extractions and how much discrepancy the pDNA extraction introduced, i.e. the higher the average ACN, the higher the discrepancy. Thus, the discrepancies observed were consistent, indicating that the pDNA extractions systematically introduced a bias that increased as the average ACN increased. This observation was added to the results section (L201-206).

Adding to this we clarified in the discussion the impact that this observed discrepancy might have on the method, propose possible explanations of its origin, and suggest possible approaches to minimize its impact on the results (L373-393).

3. For the analysis of copy number vs TZP-resistance, it is not clear why the authors calculated the increased TZP-resistance level based on the least number of copies that allowed growth at a specific TZP concentration – that suggests a simple direct correlation between the two variables. However, the graph shown in Fig. 3B demonstrates that the relationship between copy number and level of TZP resistance is likely not a straightforward one and depends on other factors. It would be better if linear regression was performed on the entire dataset, and not just the lowest copy number, to account for other factors as well.

In Fig. 3B, the ACN measured is a continuous variable (since it is the average ACN measure in a population), while the measured resistance level follows a discrete distribution with fixed concentrations of TZP (typically doubling of values each time). Thus, populations with different ACNs can have the same measured resistance level as long as the copy number of the resistance gene does not allow growth at the next higher-up resistance level value measured. However, the real biological resistance level of these populations with different ACNs falls somewhere in the continuum in between the measured resistance level and the next higher-up resistance level that could be measured. Among these populations with varying ACNs and a same measured resistance level, the population with the lowest ACN measured will be the population with a real resistance level closest to the measured resistance level. Thus, by using these values only, we can get a better estimate of the correlation between blaSHV GCN and the resistance level to TZP. To clarify this, we have modified the text in the Methods (L563-571).

Also, to figure out the relationship between the copy number and resistance, the ACN of individual colonies was measured by ddPCR. However, it is likely that there is already heterogeneity in ACN within a colony, given the number of divisions that are required to go from a single cell to a colony. Did the authors measure the ACN of a colony using long-read sequencing to test for population heterogeneity?

We agree with the reviewer that cultures are likely to show heterogeneity in ACN that could appear during the grown in absence of antibiotics. However, given the observed relative stability of ACNs over prolonged growth in the absence of TZP, we assumed that ACNs would remain relatively stable within the cultures grown in absence of TZP. Therefore the average ACN measured by ddPCR is expected to be very close to the ACN of the main population present in the colony used for the growth. Furthermore, our resistance level measured corresponds to the highest concentration at which >20% of the bacteria in the colony grew, and therefore is a measure of the ability of the main population to grow in presence of the antibiotics. Thus, we are confident that our analysis remains relatively unaffected by possible ACN changes in subpopulations present within the cultures. To clarify this point, we have modified the text in the Methods (L.543-547, 556-558)

Regarding ACN heterogeneity in colonies, unfortunately our Nanopore sequencing approach relying on the ligation library preparation kit requires larger amounts of pDNA than we could extract out of a single colony. While this is an interesting question, we have not considered using our approach to try answering it due to the required pDNA amount.

4. In the calculation of recombination rates, the authors estimate that conversion from ACN ≥ 2 to ACN=1 occurred due to recombination between whole amplified units.

However, this could also occur due to preferential selection against cells with higher copy numbers. How do the authors rule this out?

This is a good point and as the reviewer points out the loss of the ACN in a population in absence of antibiotic will be driven by two factors: the mechanistic loss rate of the amplifications due to recombination between copies and selection against the amplified states. As we have shown before (e.g. Nat Comm 2024 Pal and Andersson), which factor dominates depends on the specific amplification. However, in our specific model system fitness selection does not seem to be strong based on the following points: i) the amplified unit only encodes for only few genes, which minimizes the fitness cost of carrying additional copies of the amplified unit (Fig. 1A), ii) we could not experimentally detect a fitness cost even when n=27 amplified units were present (Fig. S6), and iii) we observed a relatively high stability of ACNs during prolonged growth in absence of antibiotics (Supplementary Fig. 4, 5 and 8). As a result, we believe that the overall loss of the ACN is mainly driven by the rates of recombination, while remaining rough estimates, and that they can be used in our mathematical model. We modified the text in the Methods to clarify this point (L.599-604).

This is also true for their data that shows that selection for fitter variants led to increased prevalence of populations with distinct ACNs (Fig. S8). If the tandem copies in a cell were recombining to alter the ACN, any increased prevalence of a specific ACN (containing the fitter variants) would be quickly lost as the ACN changed within the population.

The investigation depicted in Fig. S8 was carried out following our observation of the relative stability of ACNs following prolonged growth in absence of TZP (Supplementary Fig. 4 and 5). We assumed that, given how fast media adapted mutants took over our populations, they might lead to some specific ACNs becoming more prevalent in the population if the tandem amplifications indeed revert very slowly. Because ACNs were relatively stable and reverted slowly, we as expected observed that the specific ACNs present in the media-adapted populations became dominant in the populations and were maintained over numerous generations (supplementary Fig. S8). Indeed, data in Supplementary Fig. 8 further confirms the relative stability of ACNs during growth in absence of TZP. For example, for DA79881, over 80 generations in MH broth where only a *wecH* mutation was present, the ACN reverted very slowly. For DA79882 and DA81276, stable ACN distributions were observed over 40 and 80 generations until media-adapted mutants appeared, increased in proportion in the population and affected the ACN distribution. Thus, these observations further confirm our conclusions that additional copies of the amplified unit only conferred a marginal fitness cost. We have now added this observation and described it more clearly in the text (L250-256).

5. Long read sequencing has a bias against longer reads – how do the authors normalize for such a bias (which would predominantly affect reads corresponding to higher ACNs)?

Further, in Figures 3, S4, and S5, it appears that most of the high ACNs detected are at the limit of detection – and the Supplementary Dataset shows that in several cases there was only one read that corresponded to the highest copy number. Perhaps if the bias against longer reads can be quantified, the data could be normalized to account for such a bias, but as the data stands now, it is difficult to interpret data with single reads for a lot of data-points.

The observation and remark is correct, and our bioinformatics analysis already takes into account this bias, measure it for each Nanopore sequencing performed, and adjusts the ACN frequencies measured according to the measured bias. The bias is determined in step 7 of our bioinformatics pipeline and the adjustment performed in step 14 (see also Supplementary Fig. 2). The innovative steps of our pipeline have now been better emphasized in the Method section (L720-732).

6. During the fitness measurement, strains were grown without antibiotic and growth was measured. However, the authors only measured the ACN by ddPCR before the growth assay. It is possible that the ACN decreased to similar levels across all strains due to the absence of selection, which then led to similar fitness across the various colonies. Can the authors also do ddPCR / long read sequencing after the growth to show that the ACN did not change during the experiment?

Because ACNs were relatively stable over growth for numerous generations in MH broth in the absence of antibiotics, and because our fitness measurement of exponential growth rate in MH broth in absence of antibiotics only required short incubation in MH broth, we do not expect ACNs to change substantially during our measure of fitness. We have now clarified this in the Methods (L556-558)

Further, the fitness measurements were only performed as technical replicates, with no biological replicates.

Since each fitness measurement was performed using a single colony grown in MH broth, we did not have several biological replicates available (each colony will have a different average ACN). Re-isolation of these colonies on MHA plate might have affected the average ACN enough for the reisolated colonies to become new biological samples rather than true biological replicates. However, we tried to compensate for this by performing our fitness analysis on a large number of colonies with varying ACNs, hoping to get a statistically reliable correlation measure between detected ACN and fitness. Unfortunately, due to the low cost of the amplified unit, we did not get such data, as shown in Supplementary Fig. 6.

Minor concerns:

1. In Fig. S1, only one replicate is shown. Given that the plasmid is introduced into a different strain, and has been genetically modified, it is interesting that it has a similar effect in both strains. It would thus be useful to show 3 independent replicates each.

Since the *I-SceI* restriction site introduced on the plasmid is located outside of the amplified unit, we did not expect any major impact on the tandem amplification-driven TZP HR phenotype. Rather, we would have expected that the genetic background of the isolate carrying the plasmid could impact the PAP test results, for example by moving the PAP curve towards higher or lower TZP concentrations depending on the intrinsic TZP resistance level of the isolate. Despite this possibility, data in Fig. S1, while collected using single replicates, indicates that the modified plasmid in MG1655 and the non-modified plasmid in the clinical isolate both cause similar TZP HR phenotypes. More importantly, additional PAP tests were performed with DA76595 in triplicate and confirmed the TZP HR phenotype of MG1655 carrying the modified plasmid (Fig. S3).

2. On line 111, the authors say that “sequencing of pDNA rather than of whole genomic

DNA (gDNA) increases the sequencing depth by a theoretical 41-fold” – is this just due to the relative sizes of the genome and plasmid? Some details should be provided.

Correct. This has now been clarified in the text (L118-120).

3. In Table S2, what is “sample size”?

Sample size refers to the number of Nanopore reads that was used to determine the distribution, which were compared using the Kolmogorov-Smirnov test. The title of the column in Table S2 has been changed to “No. of reads” to clarify this, and explanation added to the legend of Table S2.

4. For Figure 3C, the authors should also describe how they estimated the frequencies of mutants that should grow at the different TZP concentrations.

We apologise if that part of our methodology was unclear. We have now added in the Methods how we estimated the frequencies of mutants based on the ACN distributions (Lines 578-586).

5. Figure legends for Figures S11-15 should clarify that these data are from a model and are not experimental data.

This has now been clarified in the legends of Fig. S11-15.

Reviewer #2 (Remarks to the Author):

Summary

This is an interesting and clearly written manuscript. While other groups have used long-read nanopore sequencing to report variable cell-to-cell amplification copy numbers in heteroresistant populations, this study achieves an unprecedented level of resolution on the distribution of tandem genetic amplifications within bacterial populations. The authors accomplished this using a novel methodology. Although their approach is limited to amplifications on plasmids, this is, to the best of my knowledge, the first time these copy number variations have been observed at this scale. Notably, their simple yet clever innovation of introducing an I-SceI restriction site adjacent to the amplification site improved read coverage. As is typical of this group, their literature review is thorough and relevant to their findings.

We thank the reviewer for the positive comments.

Major comments

As noted above, the novel approach presented requires that amplifications are plasmid-borne. These authors report the dynamics of amplification copy number distribution in the presence and absence of an antibiotic; however, understanding the relevance of their findings is somewhat dependent on establishing the clinical prevalence of plasmid-borne amplifications. The authors state that “amplification-mediated HR in E. coli is frequently plasmid-associated” and that plasmid-borne amplifications are “relatively

**common among Gram-negative pathogens” but do not provide quantitative details. To enhance the manuscript, it would be beneficial to either:
Provide additional genomic analysis of clinical isolates to quantify the prevalence of this mechanism.
Reference specific, previously published studies that directly quantify its occurrence in clinical settings.**

Due to their intrinsic instability and difficult detection requiring specific analysis methods (e.g., whole genome sequencing, ddPCR or qPCR), tandem amplifications can be difficult to detect in clinical isolates. However, we added a couple of references where such amplifications were indeed detected in clinical samples where they were linked to resistance development and treatment failure. Most implication of tandem amplifications in treatment failure in patients is still indirect. We updated the text in the introduction to emphasise better the clinical relevance of tandem amplifications (L53-60).

A key claim throughout the manuscript is the ability to "resolve single cell amplification copy numbers." This assertion hinges on the assumption that the plasmid containing the amplification unit is present at a stable single copy per cell within the population. It is important for the authors to provide supporting evidence or a robust justification for this assumption. Without this, the data more accurately represent single-molecule (or single-plasmid) resolution rather than true single-cell copy number.

Our manuscript includes data about the plasmid copy number, which we determined throughout the experiments. We have now updated the text to clarify our claim regarding detection of tandem amplifications at single cell level (L179-182).

In Figure S2 and Methods, the authors describe discarding artifact hybrid reads formed during sequencing library preparation, specifically those with multiple FR1 or FR2 sequences, or blaSHV hits outside the FR1-FR2 region. Given the high sequencing depths, it's possible to capture other rare and complex structural variants alongside the tandem amplifications. The manuscript does not clearly explain how the authors distinguish between these genuine structural variations and the discarded artifact hybrid reads. Could these artifacts also affect reads containing the tandem amplification flanked by FR1 and FR2? Further clarification on the criteria used to make this distinction is needed.

These are good and relevant points. While it is true that other rearrangements involving *blaSHV* might have happen, here we focused on tandem amplifications involving the IS sequences present on each side of the amplified unit. Artifact reads represented a very small proportion of the total amount of reads going through our pipeline. However, because such artifacts often lead to the detection of higher number of potential *blaSHV* genes on the read (e.g., if multiple FR1 or FR2 are present), they will affect more strongly our detection of reads with high ACNs where they could have more impact due to the lower count of reads carrying high ACNs. Thus, to prevent this, we decided to discard all reads that fell under our criterium of artifact reads. We have clarified in the bioinformatics pipeline the steps that remove artifact reads (L726-730).

While interesting, we did not try to detect whether the reads detected as artifacts might in fact correspond to other types of genetic rearrangements than tandem amplifications, as such

analysis might be complicated, and the focus of our work was on tandem genetic amplifications. This has been clarified now in the text (L730-732).

Minor comments

Lines 97-100. Text is at times redundant. For example, is this sentence really needed: The ability to detect rare tandem amplification events using Nanopore sequencing is dependent on the sequencing depth over the area of interest and on sequencing reads being long enough to cover the whole array of amplified units. This concept was explained in detail in the Introduction.

We have removed the redundant text in the results section (L105-106)

Figure 1A and 1B would greatly benefit from the addition of scale bars. Since the method presented depends on generating reads long enough to span the amplified arrays, the reader should be able to easily understand the scale of the amplified unit and its relationship to the engineered plasmid.

We have now added a scale bar to Fig. 1A to help the reader better grasp the size of the amplified unit. We also ensured that Fig. 1B is to the correct size. Finally, we added a new panel F to Fig. 2 that depicts examples of plasmids that are carrying different sizes of arrays of amplified units.

Line 123 Typo hypothesized -> hypotesized

This has now been corrected in the text.

Reviewer #2 (Remarks on code availability):

I did not attempt to install or run the code but it appears to be well-documented and organized such that it could be a usable resource for the community.

Reviewer #5 (Remarks to the Author):

This study provides valuable insights into the molecular mechanisms of heteroresistance, a clinically relevant but still poorly understood phenomenon. The authors focus on the dynamics of tandem gene amplification of the SHV-1 β -lactamase in Escherichia coli, investigating its dynamics in a single-copy plasmid originally found a clinical isolate. To facilitate the analysis, plasmid was transferred into the laboratory strain E. coli MG1655. Using a plasmid version that was then engineered with a unique restriction site, they established a novel robust protocol that enabled high-coverage sequencing by Nanopore Long-read sequencing of the genomic region prone to tandem amplification. Following plasmid extraction and enzymatic digestion, the authors applied sequencing-by-ligation and deep nanopore long-read sequencing. Digital droplet PCR and short-read sequencing were used to compare allele copy number estimates. This study describes interesting aspects of tandem amplification dynamics and highlight the power of nanopore long-read sequencing for resolving this type of complex genomic rearrangements. Importantly, the single-copy nature of the plasmid combined with long-read technology allow single-cell resolution of amplification events.

The main limitation of this otherwise elegant protocol is its reliance on the presence of a unique restriction site in close proximity to the resistance gene undergoing tandem amplification. This requirement may impair the generalizability of the approach to other plasmids and clinical isolates where such sites are absent or not easily engineered. It would be valuable if the authors could comment on how broadly applicable they envision this strategy to be, and whether alternative approaches might overcome this limitation.

We thank the reviewer for the positive words.

We have expanded our discussion in regards to the use of I-SceI, the limitations of our approach, and how to potentially circumvent these (L319-323, 373-393).

Specific comments:

Result section:

• Please ensure that all sequencing data are deposited in a public repository and provide the corresponding accession number

The data has been deposited and the accession number has been provided in the data availability (L855).

• L102: Was the plasmid also single copy in the original clinical isolate?

Yes, the whole genome sequencing of the original isolate revealed that it was present at a single copy in the original isolate. This information has been added to the text (L108-109).

• L106: Since plasmids were extracted prior to digestion with I-SceI, what was the rationale for selecting an enzyme that does not cut the rest of the genome?

Reviewer 5 is correct that, due to sequencing of pDNA only, other restriction sites than I-SceI might have been used. However, due to the rarity of this restriction site, the use of the same I-SceI restriction site remains valid for other plasmids, making our described approach more general.

• L117: Given that sequencing depth improved by only 22% with the digestion step, have the authors attempted the method without digestion? This could make the approach more broadly applicable to clinical isolates.

We have updated our text to emphasize the relative low impact of I-SceI digestion on our results, but still allowed for the sequencing of some intact plasmids that were present in our plasmid preparations (leading to the sequencing of full-length plasmids). We also emphasize how I-SceI digestion might be more important for DNA extractions with less fragmented DNA (L319-323).

• L163: Including the number of passages alongside the number of generations (at least once in the Results section) would help the reader contextualize the data

We added this information in the text (L172).

• L243: The authors suggest that indirect resistance might explain the wide distribution observed. Could they hypothesize further on the origin of this indirect resistance? For

example, were they able to detect free SHV-1 enzyme degrading the antibiotic in the culture medium?

We have further explained the phenomenon, which has been well studied previously (L336-339). We did not specifically try to detect free SHV-1 enzyme in the culture medium.

• L180–187: The section addressing ACNs discrepancies between Nanopore sequencing of plasmid DNA and short-read WGS/ddPCR on genomic DNA is unclear. Methods of lysis and DNA extraction are compared without further details and mixed within the same sentences. Please clarify. Are these discrepancies potentially linked to plasmid length, where larger plasmids may be sub-optimally recovered using plasmid extraction kits?

This section has been clarified in the results (L187-206) and discussed further in the discussion (L379-393)

• L211: Did the authors sequence the wild-type strain after passaging in antibiotic-free media as a control?

We did not perform parallel growth of the wild type strain in antibiotic-free media as a control. Rather, we relied on previous work where multiple cultures of *E. coli* MG1655 grown for extended periods of time in laboratory media had been genotypically and phenotypically analyzed to determine whether mutations were likely media adaptations (e.g., see Knöppel A et al, *Frontiers in Microbiology* 2018, 9,756).

Method section:

• L645: The description of mathematical modeling and computer simulation methods could be moved to the Supplementary Information, as most readers (including myself) are unlikely to be able to evaluate these details.

We have decided to maintain this section in our main text as it is an important part of the method we use for interpreting the results. to our method and the interpretation of our results. We hope that the schematic figures (Fig. S9 and S10) related to our mathematical model are sufficient to visualise the concept, while further mathematical details can be found in the Methods for the interested reader.

Figures:

• A figure displaying representative reads (e.g., using IGV, -) showing tandem amplification would greatly help readers visualize the amplification on single reads.

We added a new panel to Fig. 2 (panel F) depicting examples of plasmids carrying four different arrays of amplified units at scale.

• Figure 1B: The locations of CR5 and CR4 loci are difficult to discern; could these also be indicated in Figure 1A for improved clarity?

We have now improved Fig. 1A and added the CR4 and CR5 loci, as well as a scale bar for better visualisation of the size of the region.

• **Figure 1D: Please clarify the figure legend, as the description is currently much clearer in the text.**

The legend has now been modified and is hopefully more clear (L1041-1045).

Reviewer #5 (Remarks on code availability):

I have not reviewed the code, as this falls outside my area of expertise.

ANSWERS TO REVIEWERS' COMMENTS

Manuscript title: The dynamic distribution of genetic tandem amplifications in a heteroresistant *Escherichia coli* population revealed by ultra-deep long read sequencing

Manuscript # NCOMMS-25-68311A

Corresponding author: Dr Hervé Nicoloff

Changes related to answers to the reviewers have been highlighted in the text of the manuscript.

Reviewer #1 (Remarks to the Author):

This is a revised version of a manuscript examining dynamics of gene amplification leading to antibiotic resistance, and amplified copy numbers, in the presence and absence of antibiotics. The authors have addressed almost all of my previous concerns. I have one follow-up query below, and noticed a couple of typos:

1. I had previously raised a concern about the analysis of copy number vs TZP-resistance, querying why the authors calculated the increased TZP-resistance level based on the least number of copies that allowed growth at a specific TZP concentration. The authors' response was that while ACN is a continuous variable, the measured resistance level follows a discrete distribution. Thus, the real biological resistance of the populations with different ACNs falls somewhere in the continuum between the measured resistance level and the next higher-up resistance level.

While this explanation theoretically makes sense, it doesn't perfectly fit their data. E.g. they conclude that an ACN of ~7 should result in resistance level of 12 mg/L. However, they have an isolate with higher ACN numbers whose resistance level is only 6 mg/L (Figure 3b). Does this imply that other factors also alter resistance levels? This should be mentioned.

The reviewer is correct. When performing ASTs, it is known and accepted that MIC values measured might sometimes vary by ± 1 step between independent measurements. Here, this is likely to have happen for cultures where the ACN was at the limit between causing a resistance level reaching 12 mg/L (and measured at 12 mg/L) or remaining below 12 mg/L (and measured as 6 mg/L), which is what happened for the cultures with ± 7 copies of *bla*_{TEM}. For this reason, the linear regression had been calculated excluding the data related to resistances of 12 mg/L, where the data was less precise. We have now explained this better in the Methods (L576-579).

2. Line 99: typo in 'previously'

This has now been corrected.

3. Line 569: typo in 'vary'

This has now been corrected.

Reviewer #2 (Remarks to the Author):

Overall, it is my view that the the revised manuscript adequately addresses the concerns raised by the reviewers.

We thank the reviewer for the positive words.

With respect to the comment below, I think it is more accurate to describe detection of tandem amplifications at the single molecule, rather than single cell, level. ddPCR and WGS report the average plasmid copy number in the population and I do not think you can strictly speaking rule out changes in cell-to-cell plasmid number on the basis on these data.

Reviewer2: A key claim throughout the manuscript is the ability to "resolve single cell amplification copy numbers." This assertion hinges on the assumption that the plasmid containing the amplification unit is present at a stable single copy per cell within the population. It is important for the authors to provide supporting evidence or a robust justification for this assumption. Without this, the data more accurately represent single-molecule (or single-plasmid) resolution rather than true single-cell copy number.

Authors: Our manuscript includes data about the plasmid copy number, which we determined throughout the experiments. We have now updated the text to clarify our claim regarding detection of tandem amplifications at single cell level (L179-182).

We have removed the reference to single cell level in the abstract and and modified that statement in the discussion (L306-307). We also updated our statement in the results to acknowledge that subpopulations with increased PCN cannot be excluded, and therefore that our tandem amplification copy numbers were accurately detected at single-molecule level (L179-182).

Reviewer #2 (Remarks on code availability):

I have successfully run the code and can confirm that it works as specified.

I encountered a minor issue with Snakemake; I was unable to get it to work as described in the GitHub comment: "NB: Given that you have Snakemake installed, all other dependencies will be installed and deployed automatically when you run the pipeline (see below)." As a result, I had to install the dependencies manually. This is not a critical issue, as the code ultimately ran correctly.

We have updated README and included possibility of container execution. Simply put, it adds an extra level of reproducibility – no need to use conda, all dependencies are packed inside an isolated container. This should solve the problem. We have updates the text in the manuscript (L648-650).

I also noted a '\\n' character at the end of the FASTA file in the test data, which initially caused the run to stall. This should be an easy fix for the authors.

The '\\n' character is a puzzling thing. Possibly, the reviewer had opened and closed the test file in some text editor before running the analysis and this editor added this endline character. We checked, and our test file has standard '/n' endline character.

Reviewer #5 (Remarks to the Author):

The authors have provided the additional information requested, and my previous concerns have been adequately addressed. I have no further major comments.

We thank the reviewer for the positive words.